# Effects of the SARS-CoV-2 Pandemic on $CO_2$ Emissions in the Port Areas of the Strait of Messina

Concettina Marino , Antonino Nucara * , Maria Francesca Panzera and Matilde Pietrafesa

Department of Civil, Energetic, Environmental and Material Engineering, Mediterranea University of Reggio Calabria, 89122 Reggio Calabria, Italy; concettina.marino@unirc.it (C.M.); francesca.panzera@unirc.it (M.F.P.); matilde.pietrafesa@unirc.it (M.P.)
* Correspondence: antonino.nucara@unirc.it; Tel.: +39-0965-169-3293

**Abstract:** The Strait of Messina is characterised by a significant ship flow, especially of ferries, between its two shores. The ferry services involve four harbours, located on the Sicilian and Calabrian shores. During the epidemic emergency related to the SARS-CoV-2 virus, due to the restrictions adopted to counteract the spread of the infection, a significant reduction in ferry activities and vehicle flow occurred. These circumstances made flow data, related to different actual scenarios, available and allowed the assessment of the environmental impact of the port area. Therefore, the port area became a noteworthy case study, suited to draw conclusions regarding possible future courses of action designed to curb greenhouse gas emissions in these types of settlements. In the study, in order to assess the effect of different levels of human activity on pollutant emissions, the total $CO_2$ emissions from ferry activities in two successive years, 2019 before the spread of the virus and 2020 when the epidemic was at its peak, were evaluated and compared. The EMEP/EEA methodology was used and, as a result, an overall reduction of 13.2% in $CO_{2eq}$ yearly emission rates was observed, with the major reduction of 2784 $tCO_{2eq}$ due to maritime traffic.

**Keywords:** SARS-CoV-2 pandemic; $CO_2$ emissions; maritime traffic; road traffic; Strait of Messina; port area GWP

## 1. Introduction

In recent years, world economies have experienced various crises such as the COVID-19 pandemic, economic recessions, and increases in inflation caused by the ongoing war between Russia and Ukraine. These crises have demonstrated the fragility of European economies and their high dependence on fossil fuels.

COVID-19 has proved to be the most serious epidemiological emergency in the last century, with serious public health repercussions. The respiratory infection was identified in Wuhan city, China, in late December 2019 and, to limit its spread, the Chinese government imposed a block in the activities in Wuhan starting in January 2020 [1,2].

However, the viral infection has rapidly spread across the globe, and was declared as a public health emergency of international concern by the World Health Organization in January 2020 [3].

Since then, restrictions on travel and the suspension of all transportation have been introduced in many states to reduce personal contact and effectively halt the spread of COVID-19 [4–6] that has generated impacts in 216 countries overall.

Major international and domestic flights were cancelled; transportation systems including road, ship and rail services, with the exception of freight trains and emergency vehicles, were suspended [3,7].

As a result of the lockdown, there has been an improvement in air quality [8–10] linked to the reduced demand for transportation and to the consequent massive decrease in the combustion of fossil fuels [11,12] and reduction in $CO_2$ emissions [13].

As several studies have stated that outdoor air pollution will become the leading environmental cause of premature death in the coming decades [14–16], the need arises to analyse integrated urban areas such as port districts in relation to this issue [17–19]. There is also a need to consider the impact of such areas in terms of increasing $CO_2$ emissions and thus the greenhouse effect [20–24].

In order to limit energy dependence and simultaneously promote more efficient sustainable development by migrating toward the ecological transition, it is essential to assess the effects of policy choices regarding the reductions of carbon emissions [25–29] and to carry out analyses on actual data of strongly reduced mobility, such as that which occurred during the COVID 19 pandemic which, therefore, is an opportunity to be seized.

In this context, it is important to analyse the most emissive sectors, such as transportation [30]. Since the measures adopted in response to the recent coronavirus pandemic have significantly reduced traffic in various modalities, it is useful to analyse the effects of these changes on $CO_2$ emissions in order to appropriately address future political choices that may involve mobility.

From the point of view of air pollutant emissions, port areas play a crucial role because they involve numerous emission sources. Therefore, port area planning is deemed to be crucial in order to achieve the objectives of the European strategies aimed at energy source rational exploitation and climate change mitigation [31–33].

As a matter of fact, among the measures meant to foster decarbonisation processes, several European directives address transportation systems, with particular reference to port areas and the maritime sector [34–39].

In addition, as a general rule, port areas are located near or within urban contexts, thereby contributing to the spoiling impacts that these settlements exert on the environment. Being responsible for a large share of both European energy consumption and environmental pollution, urban areas are also one of the main topics of the European strategies concerning sustainability [40–43].

Therefore, in this framework, with a view to designing schemes addressing the effect and the consequences of global warming, the sustainable development of port areas is a crucial issue, which needs the development of appropriate planning procedures and strategies.

In order to provide a suitable contribution to this matter, the present study tries to test a feasible procedure to assess the impact of port areas in terms of greenhouse gas emissions generated by their various activities, which are typical of these facilities.

The subject of the analysis is the port system of the Strait of Messina, a southern Italy area characterized by high ferry activities. It is appropriate to test the suitability of the present study's procedures since it is located within a large metropolitan area, which extends on both side of the Strait and has about 800.000 inhabitants [44]. Therefore, the contribution of port activities in a huge urban context can be assessed.

In addition, the selected port system is composed of four harbours characterized by different dimensions and activities: Messina, Tremestieri, Villa San Giovanni and Reggio Calabria. These circumstances allow the analysis to be carried out from two points of view: the impacts of different contexts, with different dimensions; and the feasibility of the procedure in assessing various settings.

As far as the time frame is concerned, greenhouse gas emissions discharged by the activities performed within the four port areas were assessed on a yearly basis. Specifically, two years were considered: 2019 and 2020. The rationale of this choice regards the possibility of examining the effect of diverse activity levels on the yearly emission rates.

As a matter of fact, in January 2020, in Italy, the pandemic emergency status was declared [45] and in March 2020, lockdown was instituted [46]. This caused a strong reduction in "non-vital" activities and, consequently, in transportation demand.

Therefore, the effect of this unique circumstance can be used for comparison purposes and the paper focuses on the results of this analysis in order to identify possible priorities and interventions to be implemented with the aim of reducing greenhouse gas emissions.

## 2. Methodology

The methodology used to estimate emissions is based on the EMEP/EEA air pollutant emission inventory guidebook 2019 [47], deriving from the IPCC Guidelines for National Greenhouse Gas Inventories [48]. As a matter of fact, this methodology allows the assessment of the contribution of several pollutant source sectors, such as navigation, rail, road traffic, etc., whereas other methods may be used for the evaluation of a single sector only [49–56].

The ports' emissions due to vessel traffic were determined within the port area according to the power of the vessels' engines and the time required for the phases of docking, hotelling and departure within the port; emissions from road traffic were also determined for each category of vehicle according to the distance from the docking area to the nearest boundary section of the study area; and emissions due to rail traffic were evaluated based on fuel consumption.

### 2.1. Emissions by Maritime Traffic

To obtain the pollutant emissions due to maritime traffic, it was necessary to determine the ship's energy consumption in the port during docking and departure manoeuvres, and in the hotelling phase.

The calculation was conducted by referring to the EMEP/EEA methodology related to navigation [57].

According to the methodology, total emissions, expressed in tons of pollutants, are determined with the following equation:

$$E^k = \sum_{j=1}^{N_c} CE_j FE_j^k \qquad (1)$$

where:

- $CE$, total energy consumption (MWh);
- $FE$, pollutant emission factor (t/MWh);
- $j$, fuel;
- $k$, pollutant;
- $N_c$, number of used fuels.

Instead, the total energy consumption with reference to a specific period for the *j*-th fuel used, in MWh, is equal to:

$$CE_j = 10^{-3} N_j \sum_{n=1}^{N_n} \sum_{f=1}^{4} CE_{n,f} \qquad (2)$$

where:

- $N$, number of ships in the time period;
- $j$, fuel;
- $f$, phase (docking, hotelling, departure);
- $n$, vessel type;
- $N_n$, number of vessel types.

In the previous equation, the energy consumption of a single vessel type, per phase, expressed in kWh/phase, is equal to:

$$CE_{n,f} = P_n t_{n,f} \left( L_{n,f,MAIN} FT_f + R_{A/M} L_{n,f,AUX} \right) \qquad (3)$$

where:

- $P$, maximum engine power (kW);
- $t$, time (hours);

- *L*, load factor, given by ratio of the power engaged in the phase to the maximum engine power;
- *FT*, percentage of operating time of the main engine;
- $R_{A/M}$, ratio of auxiliary engine power to main engine power;
- *n*, vessel type;
- *f*, phase (docking, hotelling, departure);
- *MAIN*, main engine;
- *AUX*, auxiliary engine.

The load factors, *L*, for the main and auxiliary engines were assumed with reference to Table 1. The emission factor in terms $CO_{2eq}$ was assumed to be equal to 0.268 t/MWh [58].

**Table 1.** Load factors by engine type and phase.

| Engine Type | Docking | Hotelling | Departure |
|:---:|:---:|:---:|:---:|
| Main | 0.20 | 0.20 | 0.20 |
| Auxiliary | 0.50 | 0.40 | 0.50 |

*2.2. Emissions by Road Traffic*

The calculation of road traffic emissions was conducted with reference to the methodology given in the EMEP/EEA guidelines for road transport emissions [59].

The methodological approach for estimating pollutant emissions by road traffic involves different levels of detail, from Tier 1 to Tier 3, and in general the variability of traffic parameters, such as vehicle speed and driving modes, should be considered [60].

The choice of the approach is related to data availability and, in this case, the analysis was conducted using Tier 1; so, the total emissions for each pollutant, in tons, were calculated through the following expression:

$$E^k = 10^{-6} \sum_{v=1}^{N_v} N_v d_v FE_v^k \qquad (4)$$

where:

- *N*, number of vehicles (vehicle);
- *d*, average distance travelled in the port area by vehicles (km/vehicle);
- *FE*, pollutant emission factor (g/km);
- *v*, vehicle category (cars, trucks);
- *k*, pollutant;
- $N_v$, number of vehicle categories.

The emission factor in terms of $CO_{2eq}$ for each vehicle category was obtained from the emission factors of single pollutants with the relationship:

$$FE_v^{CO_{2eq}} = \sum_{k=1}^{N} GWP^k FE_v^k \qquad (5)$$

in which:

- *GWP*, Global Warming Potential;
- *FE*, pollutant emission factor (g/km);
- *v*, vehicle category (cars, trucks);
- *k*, pollutant.

Emission factors for the considered greenhouse gases ($CO_2$, $CH_4$, $N_2O$) for each vehicle category were obtained from the database of average emission factors for road transport in Italy given by the "Istituto Superiore per la Protezione e la Ricerca Ambientale" (ISPRA) [61]. Furthermore, the values of Global Warming Potentials were derived from the

Climate Change 2007 IPCC Synthesis Report [62] for a 100-year time period. The $CO_{2eq}$ values obtained were 978.69 g/km for heavy trucks and 248.04 g/km for cars.

*2.3. Emissions by Rail Traffic*

Pollutant emissions from rail traffic are due to the movement of rail wagons from railway stations to ferries. Referring to the EMEP/EEA methodology related to rail traffic [63], the calculation of emissions, in tons, was conducted using the following relation:

$$E^k = 10^{-3} \sum_{j=1}^{N_c} CC_j FE_j^k \tag{6}$$

where:

- $CC$, fuel consumption (t);
- $FE$, pollutant emission factor (kg/t);
- $j$, fuel;
- $k$, pollutant;
- $N_c$, number of fuels.

Assuming the wagons are moved by diesel shunting locomotives, according to the data provided by [63] and with reference to the pollutants $CO_2$, $CH_4$, $N_2O$, the emission factor in terms of $CO_{2eq}$ was 3195 kg/t.

**3. Case Study**

Using the methodology previously described, it was possible to assess the effect on $CO_2$ emissions provoked by the changing conditions in ferry traffic between the ports of the Strait of Messina due to the spread of the COVID-19 pandemic.

The emissions generated in each port area and produced by ship traffic, road vehicular traffic, and rail traffic were evaluated.

In order to proceed with the calculation of emissions, it was necessary to know data about ferry traffic, vehicular flow in the port areas, the composition of the vessel fleet, distinguished by passenger, ro-ro and railway ships, the composition of the vehicular fleet, distinguished in the two categories of cars and heavy-duty vehicles, and the knowledge of the emission factors of the different pollutants.

The analysis was conducted referring to the years 2019 and 2020 in the areas of the ports of Messina (ME), Reggio Calabria (RC), Tremestieri (TREM) and Villa San Giovanni (VSG).

The procedure was developed through the following steps:

1. Analysis of vessel and vehicular flows between the ports on both sides of the Strait of Messina;
2. Identification of the characteristics of the ships' engines;
3. Determination of the times spent in each port for each vessel typology in the docking, hotelling and departure phases;
4. Determination of the energy consumption and emissions of vessel for the ferry service on the Strait of Messina;
5. Determination of vehicle paths within each port area;
6. Determination of energy consumption and pollutant emissions of the vehicle fleet in the areas in the proximity of each port;
7. Determination of the energy consumption and pollutant emissions produced by the shunting locomotives used for the movement of railway wagons ferrying across the Strait of Messina;
8. Determination of overall emissions in each port area.

*3.1. Territorial Organization*

The initial phase of the work was focused on defining the study area.

The territorial area in which the analysis was carried out includes the Strait of Messina, bounded by the ports of Messina and Tremestieri on the Sicilian coast and by the ports of Reggio Calabria and Villa San Giovanni on the Calabrian coast (Figure 1).

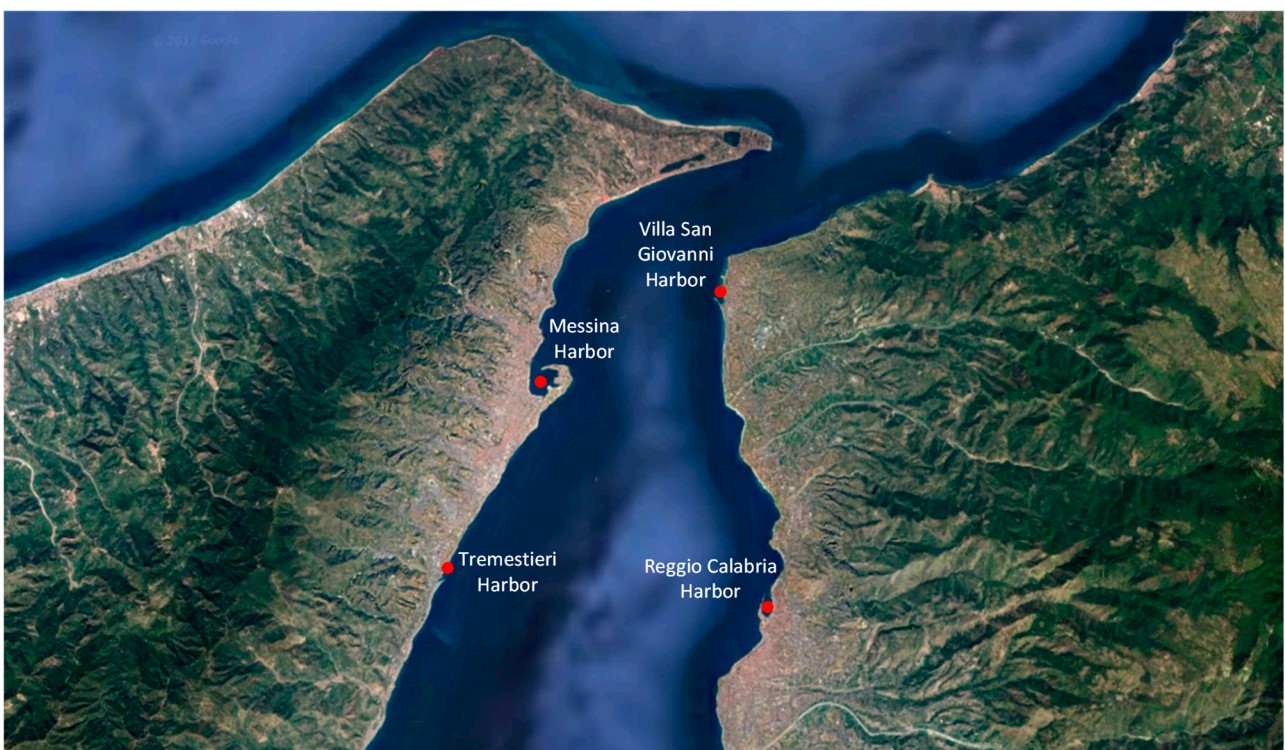

**Figure 1.** Study area: Strait of Messina (Source: Map data © 2023 Google).

The Strait of Messina area is part of the territory of the provinces of Messina and Reggio Calabria, which together have a population of about 1,100,000, of which 603,980 are in the province of Messina and 523,791 in the province of Reggio Calabria [64].

To identify the area, reference is often made to the Strait metropolitan area, including the urban centres of Messina, Villa San Giovanni and Reggio Calabria or, more extensively, to the area between Melito di Porto Salvo and Bagnara Calabra on the Calabrian shore and the area between Milazzo, with the Aeolian Islands archipelago, and Taormina for the Sicilian shore.

The Strait's metropolitan area thus defined results in a population of about 800,000, with a central area (Reggio–Messina–Villa S. Giovanni) of 437,500, comparable to other European metropolitan areas.

More than 30% of the demand for mobility between the two sides of the Strait is constituted by residents of the municipalities belonging to the area. An analysis of the origins and destinations of these movements shows a prevalence of transfers from Calabria to Sicily for study and from Sicily to Calabria for work. The remaining movements concern flows between the centres of Sicily and the continent, at medium and long distances [44].

In this study, the sailing routes between the ports were identified, specifying the type of ship serving each shipping line (Figure 2); then, individual port areas were identified, with the seaward boundary located at the mouth of the port and the landward boundary corresponding to the section of the highway junction closest to the port.

### 3.2. Maritime Traffic

Data of ferry flows between the shores of the Strait of Messina were taken from the Strait Port System Energy and Environmental Planning Document produced by the Strait Port System Authority [44].

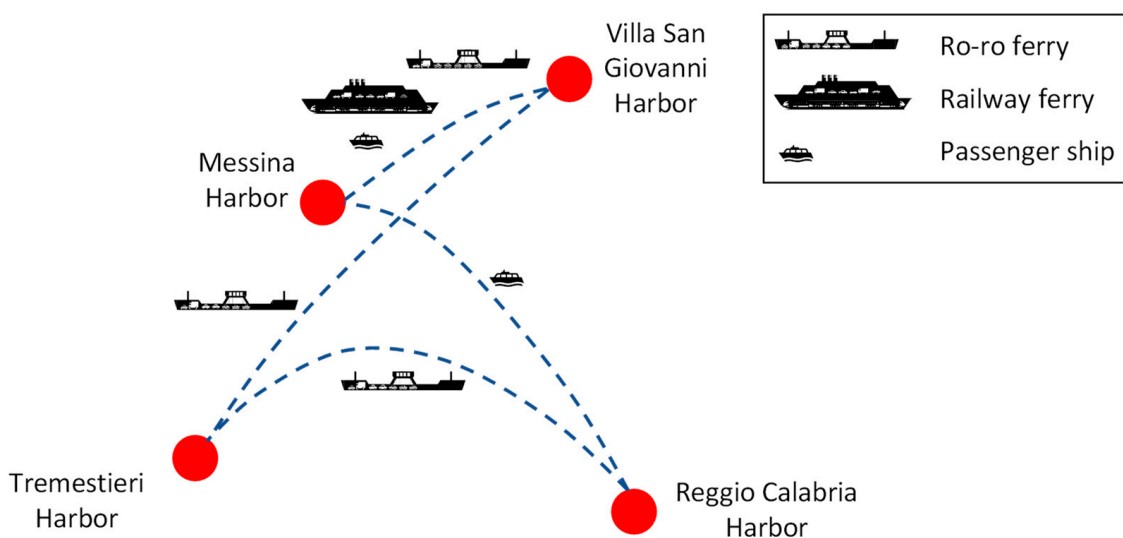

**Figure 2.** Sailing routes between ports by ship type.

The annual summary data on ferry service are shown in Figure 3.

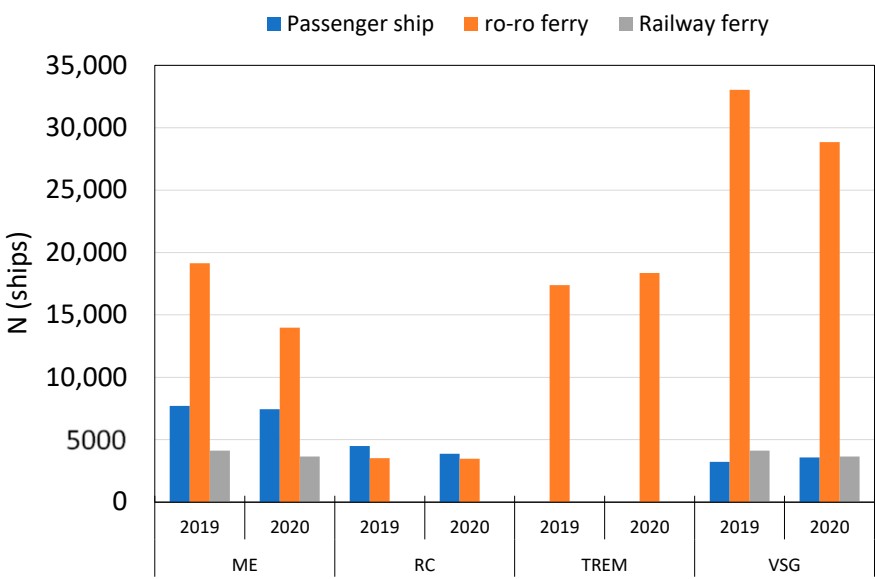

**Figure 3.** Annual ship movements in the ports of the Strait of Messina.

Detailed data with the total number of vessel trips between origin and destination ports in the years 2019 and 2020 for the three strait ferrying modes are shown in the following tables (Tables 2–7), where the rows identify the origins of movement and the columns identify the destinations.

**Table 2.** Passenger ships movements 2019.

| O/D | ME | RC | TREM | VSG |
|---|---|---|---|---|
| ME | - | 4492 | 0 | 3215 |
| RC | 4492 | - | 0 | 0 |
| TREM | 0 | 0 | - | 0 |
| VSG | 3215 | 0 | 0 | - |

**Table 3.** Passenger ships movements 2020.

| O/D | ME | RC | TREM | VSG |
|---|---|---|---|---|
| ME | - | 3865 | 0 | 3572 |
| RC | 3865 | - | 0 | 0 |
| TREM | 0 | 0 | - | 0 |
| VSG | 3572 | 0 | 0 | - |

**Table 4.** Ro-ro ferries movements 2019.

| O/D | ME | RC | TREM | VSG |
|---|---|---|---|---|
| ME | - | 0 | 0 | 19,150 |
| RC | 0 | - | 3509 | 0 |
| TREM | 0 | 3509 | - | 13,882 |
| VSG | 19,150 | 0 | 13,882 | - |

**Table 5.** Ro-ro ferries movements 2020.

| O/D | ME | RC | TREM | VSG |
|---|---|---|---|---|
| ME | - | 0 | 0 | 13,970 |
| RC | 0 | - | 3467 | 0 |
| TREM | 0 | 3467 | - | 14,886 |
| VSG | 13,970 | 0 | 14,886 | - |

**Table 6.** Railway ferries movements 2019.

| O/D | ME | RC | TREM | VSG |
|---|---|---|---|---|
| ME | - | 0 | 0 | 4117 |
| RC | 0 | - | 0 | 0 |
| TREM | 0 | 0 | - | 0 |
| VSG | 4117 | 0 | 0 | - |

**Table 7.** Railway ferries movements 2020.

| O/D | ME | RC | TREM | VSG |
|---|---|---|---|---|
| ME | - | 0 | 0 | 3644 |
| RC | 0 | - | 0 | 0 |
| TREM | 0 | 0 | - | 0 |
| VSG | 3644 | 0 | 0 | - |

It can be seen that in all ports there was an annual reduction in ship traffic. Only moderate increases in the number of ro-ro ferries at the port of Tremestieri and in the number of passenger ships at the port of VSG were noted.

Instead, the following figures show the monthly ferry departures from each port for ro-ro ferries (Figure 4), railway ferries (Figure 5) and passenger ships (Figure 6).

On a monthly basis, in the first months of 2020, and particularly in April, a significant reduction in ro-ro (Figure 4) and railway ferries (Figure 5) is noted at the ports of Messina and Villa San Giovanni; on the contrary, no significant differences in the number of ro-ro ferries are shown at the Port of Reggio Calabria, while there was a slight increase in June, July and November at the port of Tremestieri.

In all ports affected by passenger traffic, a significant reduction in the number of ships was observed in April 2020, followed by an increase beginning in August for the ports of Messina and Villa San Giovanni (Figure 6).

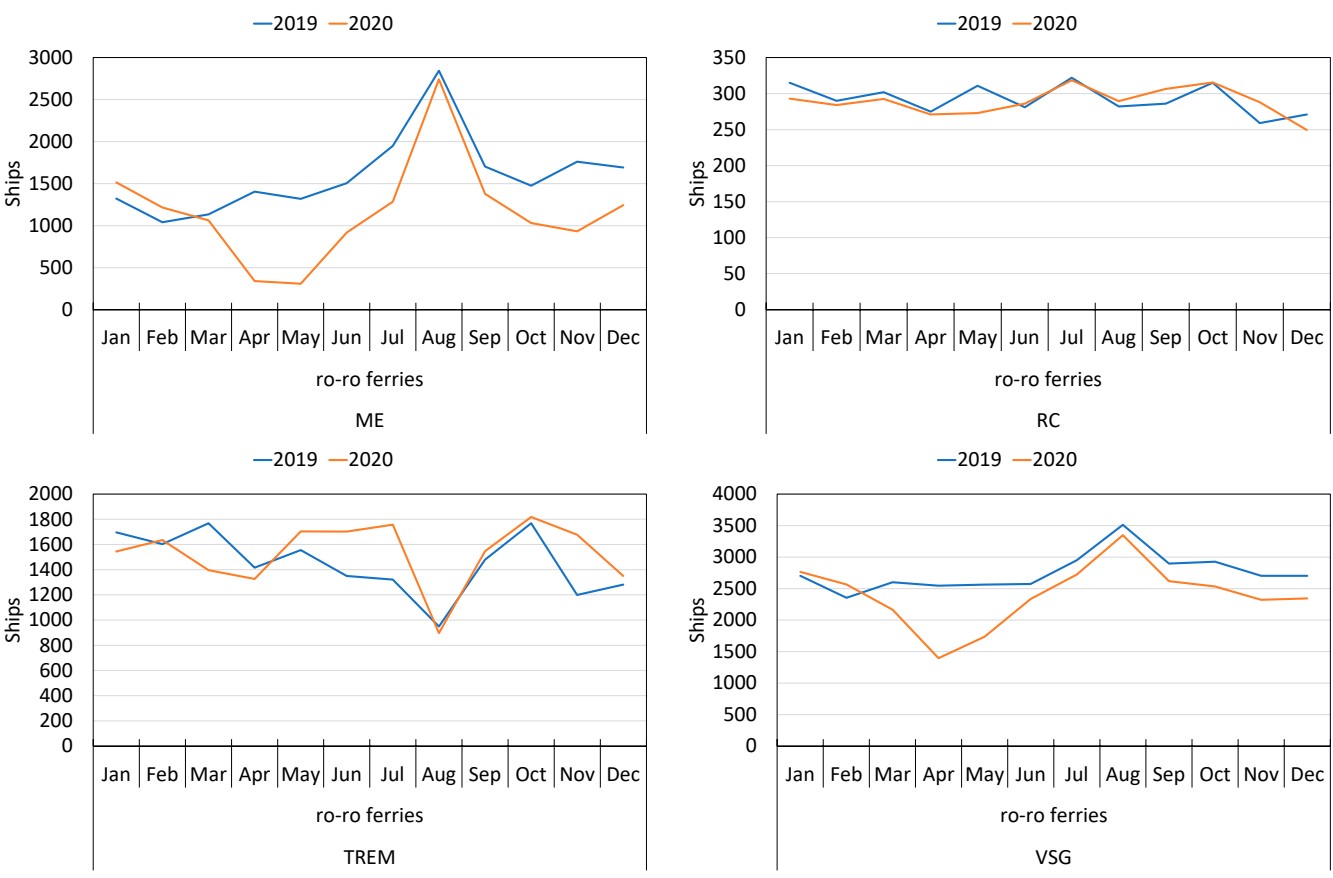

**Figure 4.** Monthly ro-ro ferry movements.

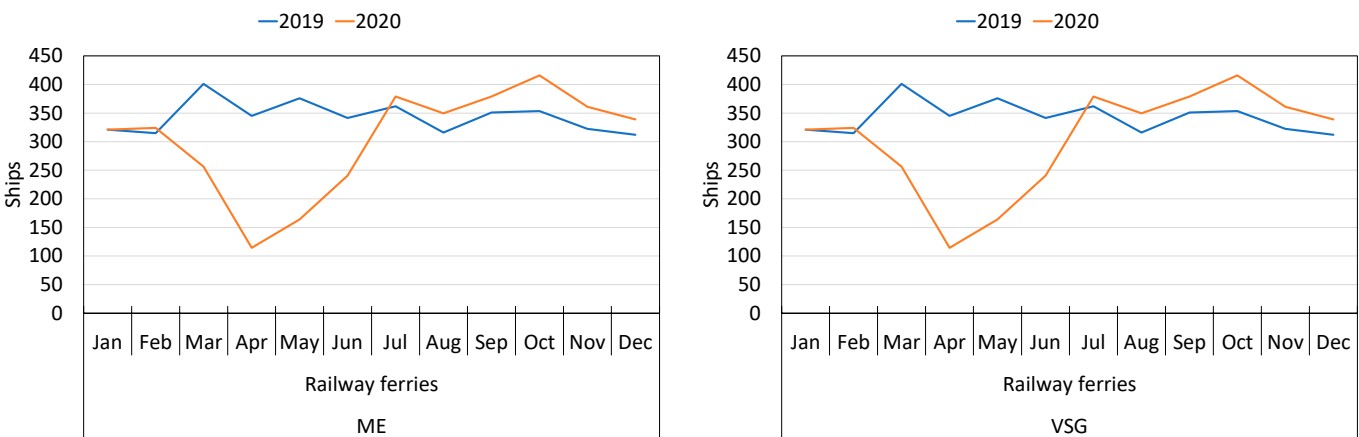

**Figure 5.** Monthly railway ferry movements.

*3.3. Road Traffic*

The total number of vehicles which travelled (boarding and de-boarding) between the ports of origin and destination in the years 2019 and 2020, distinguished into passenger cars and heavy-duty trucks, is shown in the following Tables (Tables 8–11).

With reference to the year 2020, a reduction in vehicular traffic volumes compared to the previous year is noticeable in all ports with the exception of the port of Tremestieri where, during the pandemic, some of the traffic that led to Messina was transferred.

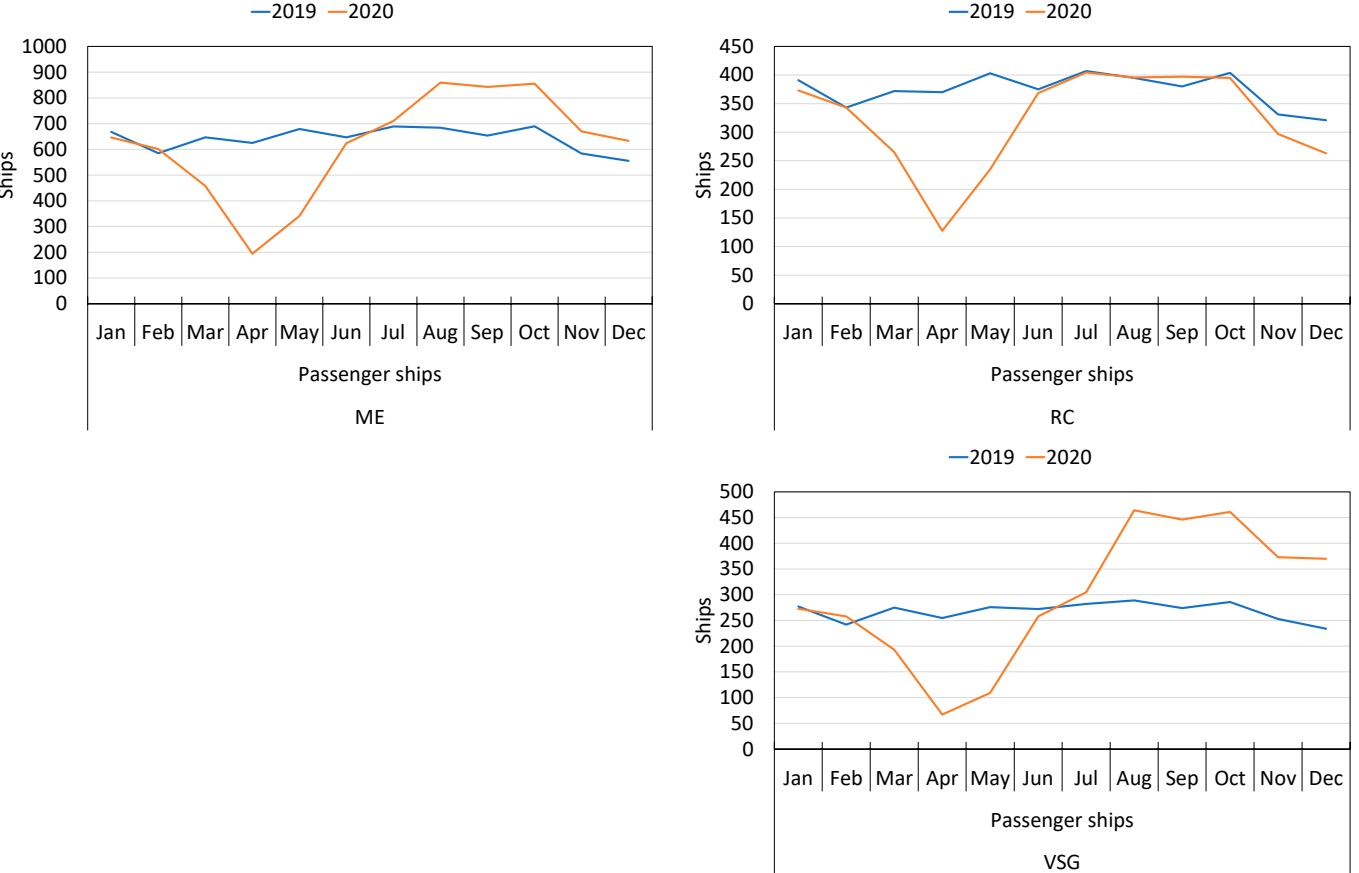

**Figure 6.** Monthly passenger ship movements.

**Table 8.** Passengers car movements 2019.

| O/D | ME | RC | TREM | VSG |
|---|---|---|---|---|
| ME | | | | 1,767,513 |
| RC | | | 8282 | |
| TREM | | 8282 | | 18,832 |
| VSG | 1,767,513 | | 18,832 | |

**Table 9.** Passengers car movements 2020.

| O/D | ME | RC | TREM | VSG |
|---|---|---|---|---|
| ME | | | | 1,301,434 |
| RC | | | 4612 | |
| TREM | | 4612 | | 34,851 |
| VSG | 1,301,434 | | 34,851 | |

**Table 10.** Heavy-duty truck movements 2019.

| O/D | ME | RC | TREM | VSG |
|---|---|---|---|---|
| ME | | | | 315,279 |
| RC | | | 98,960 | |
| TREM | | 98,960 | | 354,815 |
| VSG | 315,279 | | 354,815 | |

**Table 11.** Heavy-duty truck movements 2020.

| O/D | ME | RC | TREM | VSG |
|---|---|---|---|---|
| ME | | | | 207,249 |
| RC | | | 96,248 | |
| TREM | | 96,248 | | 427,474 |
| VSG | 207,249 | | 427,474 | |

Monthly transit for each port separated into cars (Figure 7) and trucks (Figure 8) are shown.

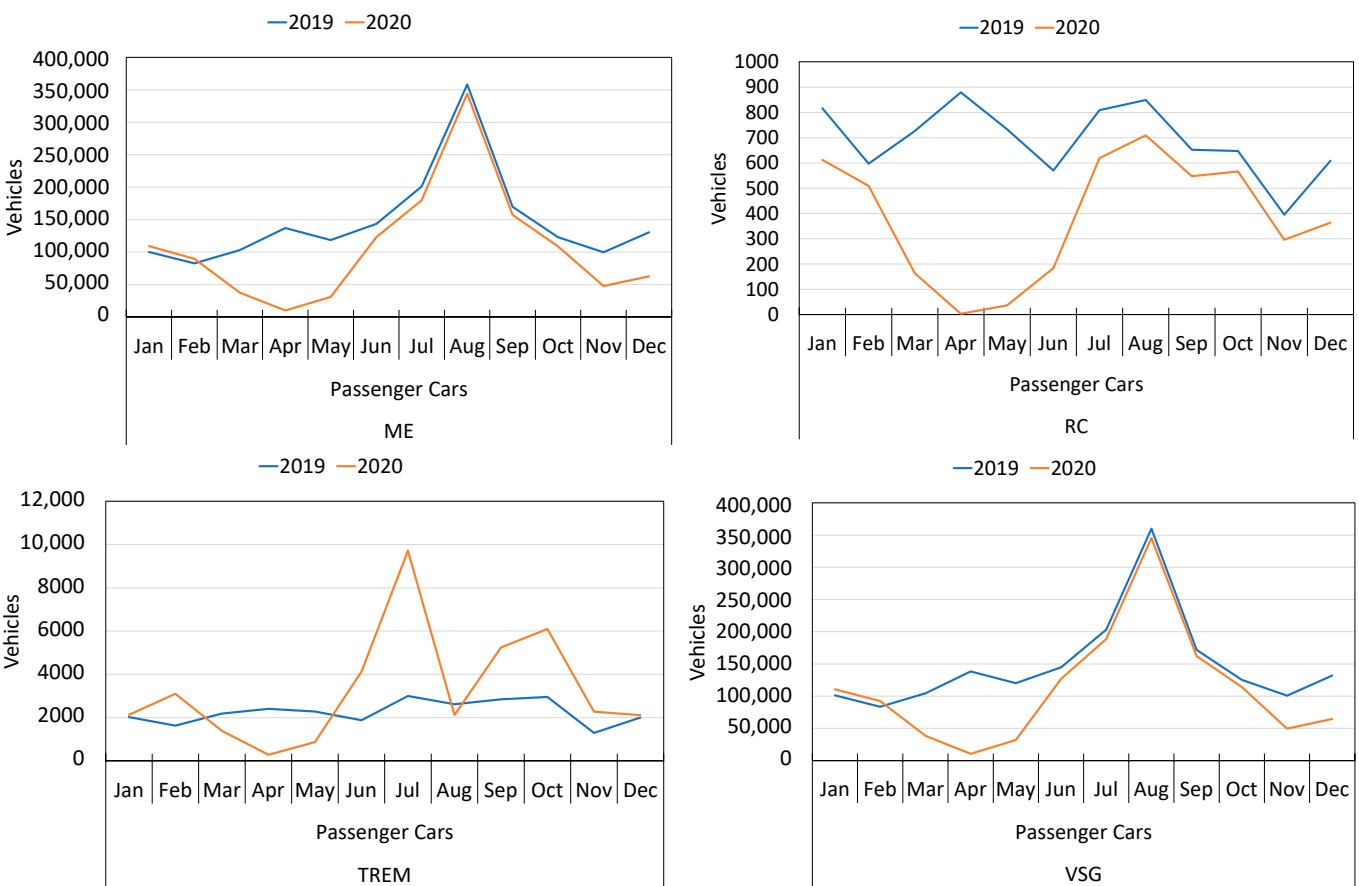

**Figure 7.** Monthly car movements.

On a monthly basis, significant reductions were seen in April and May 2020 in passenger cars. In these months, the number of vehicles was almost zero (Figure 7).

During the same period, significant reductions were also noted in the number of heavy-duty trucks at the Port of Messina, while smaller changes occurred at the ports of Reggio Calabria and Villa San Giovanni (Figure 8). Finally, due to the transfer of part of the road traffic directed to Messina to the port of Tremestieri, monthly increases in passenger cars (Figure 7) and heavy-duty vehicles (Figure 8) occurred at the port of Tremestieri.

*3.4. Rail Traffic*

The rail flows consist of all the movements of shunting locomotives through the ports of Messina and Villa San Giovanni, aimed at the boarding of railway wagons.

The total number of railway wagons moved monthly in the years 2019 and 2020, which is the same for each port, is shown in Figure 9. An annual reduction in ferried wagons in 2020 is evident, with the largest decrease in March and October.

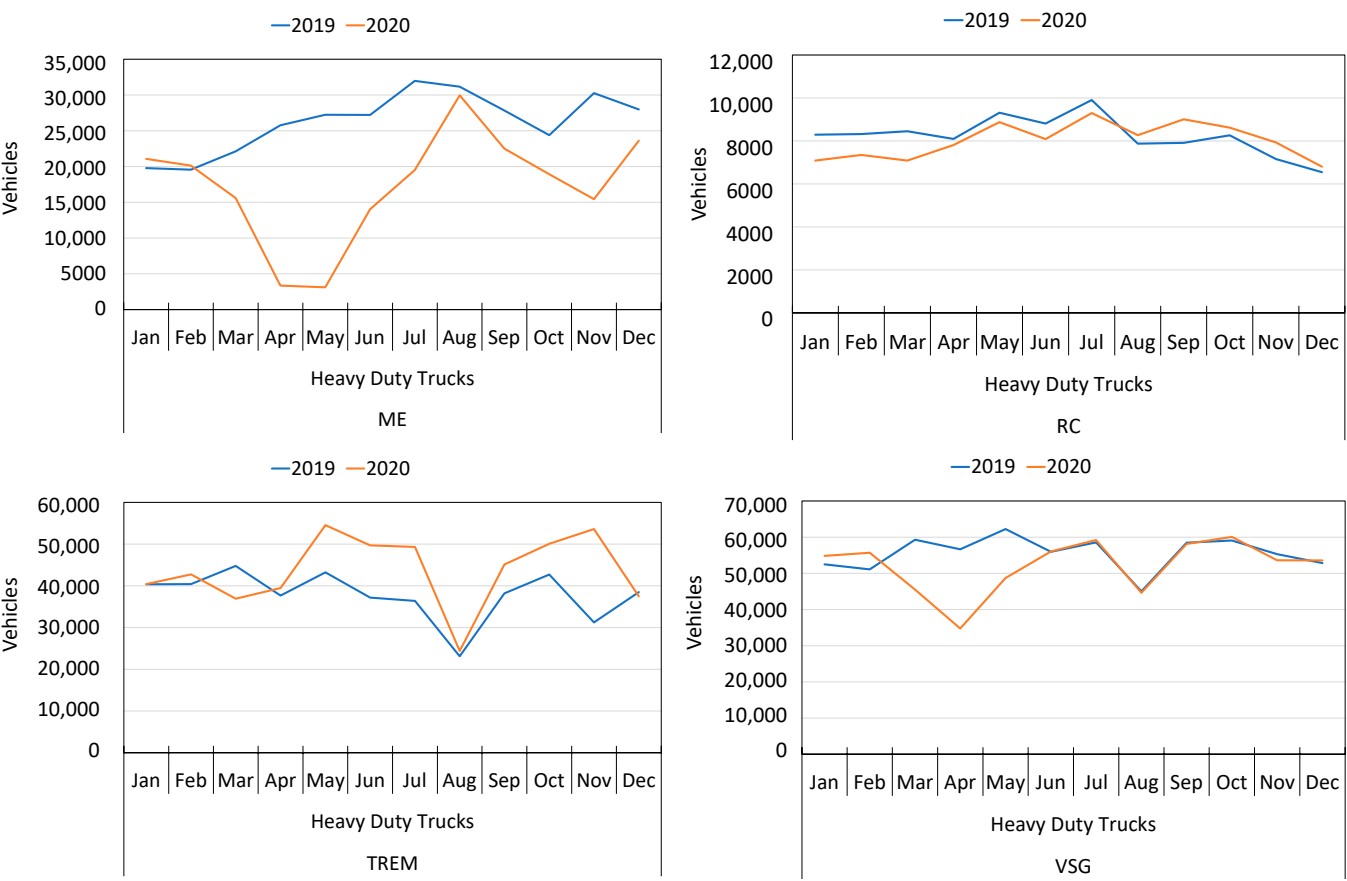

**Figure 8.** Monthly heavy-duty truck movements.

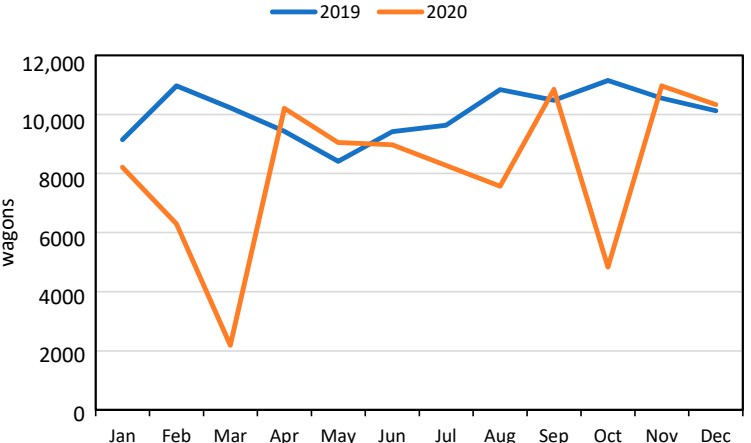

**Figure 9.** Rail wagon movements.

*3.5. Port of Messina*

The port of Messina (Figure 10) is the largest natural harbour in Sicily. Classified as Category II and Class I, it is totally docked, with a north-facing mouth about 400 m wide and a water surface of about 75 ha. The shoreside quays extend for about 1770 m, with a depth of the seabed between 6.5 m and 13 m.

The routes start from the port of Messina to the port of Villa San Giovanni for vehicle and rail traffic or reach the ports of Reggio Calabria and Villa San Giovanni for passenger traffic. Ferry service is realised with ro-ro ships for the road vehicles, with ferries for railway wagons and with motor vessels for passengers.

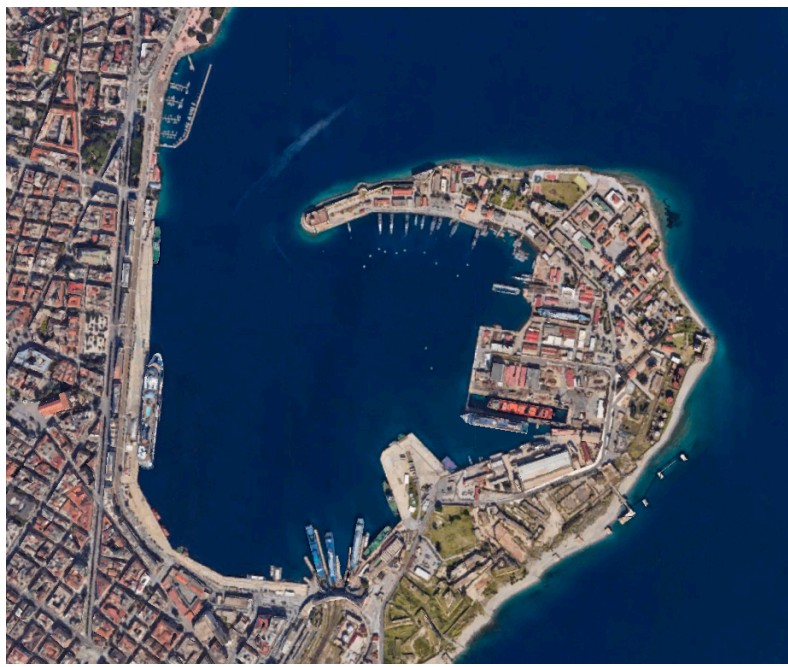

**Figure 10.** Messina port (Source: Map data © 2023 Google).

Ships' technical characteristics in terms of maximum power and ratio of auxiliary engine power to main engine power are shown in Table 12, while the durations of the single phases of docking, hotelling and departure are shown in Table 13.

**Table 12.** Characteristics of engines of ships in service at Messina port.

| Type of Ship | Company | $P_{max}$ (kW) | $P_{aux}/P_{main}$ |
|---|---|---|---|
| Passenger ship | Blu Jet | 4000 | 0.27 |
| Ro-ro ferry | Caronte &Tourist | 8800 | 0.39 |
| | Bluferries | 7900 | 0.39 |
| Railway ferry | R.F.I. | 10,500 | 0.39 |

**Table 13.** Average times for each phase at Messina port.

| Type of Ship | $t_{docking}$ (min) | $t_{hotelling}$ (min) | $t_{departure}$ (min) |
|---|---|---|---|
| Passenger ship | 3 | 10 | 3 |
| Ro-ro ferry (Caronte & Tourist) | 3 | 20 | 5 |
| Ro-ro ferry (Blueferries) | 4 | 20 | 6 |
| Railway ferry | 12 | 60 | 7 |

Emissions deriving from vehicle traffic are due to heavy-duty trucks and cars directed toward the port of Villa San Giovanni and coming from the same port in the journey from the landing dock to the "Messina—Boccetta" motorway junction.

Distances between ferry terminals and the motorway junction are reported in Table 14.

**Table 14.** Distances Ferry Terminal—Messina–Boccetta motorway junction.

| Terminal | Terminal → Junction $d_v$ (km) | Junction → Terminal $d_v$ (km) |
|---|---|---|
| Blufferies | 3.6 | 3.6 |
| Caronte & Tourist | 3.5 | 4.2 |

Railway emissions for ferry service through the Strait are due to the movements of the shunting locomotives used for boarding railway wagons during the trip from Messina Marittima station to the ferry docks.

### 3.6. Port of Reggio Calabria

The port of Reggio Calabria (Figure 11), classified as a Category II and Class II port, consists of an artificial basin protected by a wharf on the west. On the inner side, there is the Margottini Quay and, further to the south, the pier breakwater for pleasure craft.

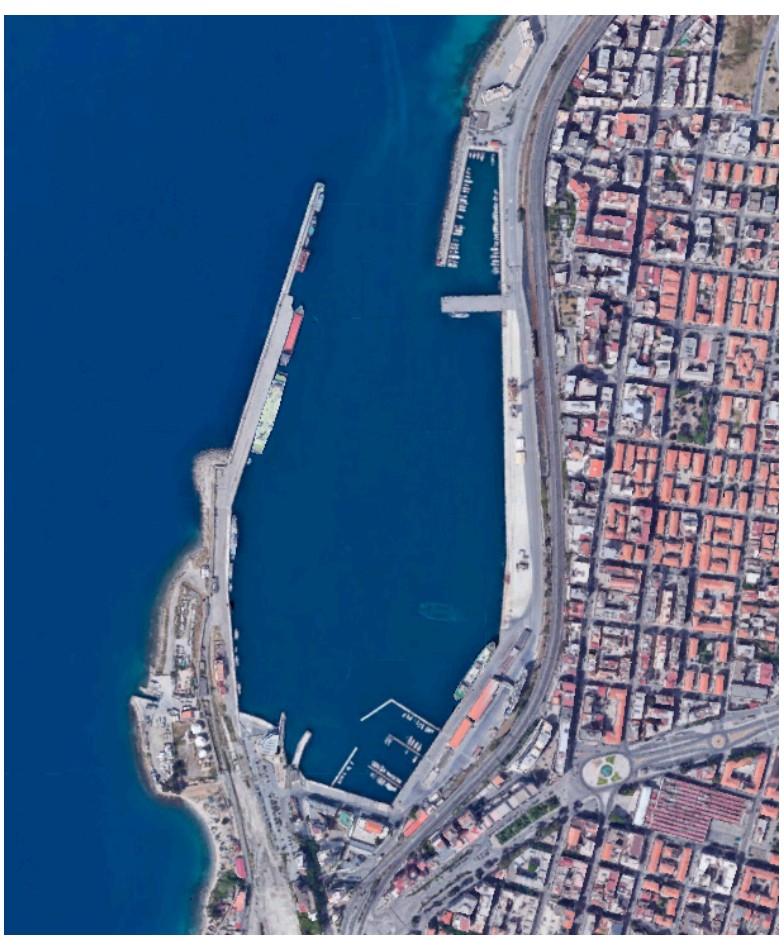

**Figure 11.** Reggio Calabria port (Source: Map data © 2023 Google).

The mouth is 110 m wide, while the quays measure approximately 2.5 km. The seabed has an average depth of 7.50 m, while the total area of the basin occupies approximately 10 ha.

Routes departing from the port of Reggio Calabria reach the port of Tremestieri for vehicle transport and the port of Messina for passenger traffic. Ferry service is realised with ro-ro ships for the road vehicles and with motor vessels for passengers.

Ships' technical characteristics in terms of maximum power and ratio between auxiliary engine power and main engine power are shown in Table 15, while the durations of the single phases of docking, hotelling and departure are shown in Table 16.

**Table 15.** Characteristics of engines of ships in service at Reggio Calabria port.

| Type of Ship | Company | $P_{max}$ (kW) | $P_{aux}/P_{main}$ |
|---|---|---|---|
| Passenger ship | Blu Jet | 4000 | 0.27 |
| Ro-ro ferry | Meridiano | 3678 | 0.39 |

**Table 16.** Average times for each phase at Reggio Calabria port.

| Type of Ship | $t_{docking}$ (min) | $t_{hotelling}$ (min) | $t_{departure}$ (min) |
|---|---|---|---|
| Passenger ship | 3 | 10 | 3 |
| Ro-ro ferry | 6 | 20 | 4 |

Emissions from vehicular traffic are due to heavy-duty trucks and cars directed toward the port of Tremestieri and coming from the same port, in the itinerary from the landing dock to the "Reggio Calabria—Porto" motorway junction. Distances between ferry terminals and the motorway junction are reported in Table 17.

**Table 17.** Distances Ferry Terminal—Reggio Calabria–Porto motorway junction.

| Terminal | Terminal → Junction $d_v$ (km) | Junction → Terminal $d_v$ (km) |
|---|---|---|
| Meridiano | 3.1 | 2.6 |

### 3.7. Port of Tremestieri

The port of Tremestieri (Figure 12), built with the aim of rerouting part of the traffic from Messina's historic centre, was completed in 2006. It is an artificial basin, protected by an east-facing breakwater at the base of which the docks for the embarking of road vehicles are located. The mouth is 80 m wide, while the quaysides measure approximately 400 m. The seabed is sandy and between 5 and 10 m deep, while the total area of the basin occupies approximately 1.3 ha.

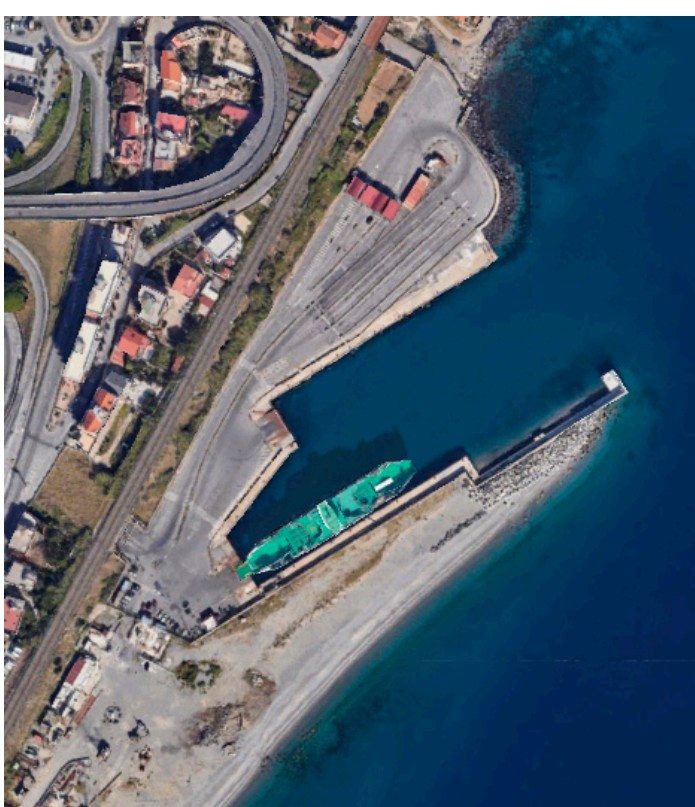

**Figure 12.** Tremestieri port (Source: Map data © 2023 Google).

Routes departing from the port of Tremestieri have as destinations the ports of Reggio Calabria and Villa San Giovanni; only road vehicles are ferried, and they are ferried by ro-ro ships.

Ships' technical characteristics in terms of maximum power and ratio of auxiliary engine power to main engine power are shown in Table 18, while the durations of the single phases of docking, hotelling and departure are shown in Table 19.

**Table 18.** Characteristics of engines of ships in service at Tremestieri port.

| Type of Ship | Company | $P_{max}$ (kW) | $P_{aux}/P_{main}$ |
|---|---|---|---|
| | Meridiano | 3678 | 0.39 |
| Ro-ro ferry | Caronte &Tourist | 8800 | 0.39 |
| | Bluferries | 7900 | 0.39 |

**Table 19.** Average time for each phase at Tremestieri port.

| Type of Ship | $t_{docking}$ (min) | $t_{hotelling}$ (min) | $t_{departure}$ (min) |
|---|---|---|---|
| Ro-ro ferry | 5 | 20 | 3 |

Emissions from vehicle traffic are due to heavy-duty trucks and cars directed toward the ports of Villa San Giovanni and Reggio Calabria and coming from the same ports, in the itinerary from the landing dock to the "Messina—Tremestieri" motorway junction. Distances between ferry terminals and the motorway junction are reported in Table 20.

**Table 20.** Distances Ferry Terminal—Messina–Tremestieri motorway junction.

| Terminal | Terminal → Junction $d_v$ (km) | Junction → Terminal $d_v$ (km) |
|---|---|---|
| Meridiano Blufferies Caronte & Tourist | 3.1 | 2.6 |

### 3.8. Port of Villa San Giovanni

The port of Villa S. Giovanni (Figure 13), classified as a Category II and Class II port, is an artificial basin protected by a straight breakwater at the base of which there are railroad docks: of these, three are specialized for rail ferry service and one for road-vehicle embarking. The dock extends northward and is used for the landing of ferries that operate for ferry passengers, cars and heavy-duty vehicles along the routes to Messina and Tremestieri. The mouth is 230 m wide and the quays measure about 1.2 km. The seabed is sandy and between 5 and 10 m deep. The total area of the basin occupies around 6.7 ha.

The routes departing from the port of Villa San Giovanni reach the port of Messina and Tremestieri for vehicle and rail traffic, and the ports of Messina for passenger traffic. Ferry services are realised with ro-ro ships for the road vehicles, with ferries for railway wagons and with motor vessels for passengers.

Ships' technical characteristics in terms of maximum power and ratio of auxiliary engine power to main engine power are shown in Table 21, while the durations of the single phases of docking, hotelling and departure are shown in Table 22.

Emissions from vehicle traffic are due to heavy-duty trucks and cars directed toward the ports of Messina and Tremestieri and coming from the same ports in the journey from the landing dock to the "Villa San Giovanni" motorway junction. Distances between ferry terminals and the motorway junction are reported in Table 23.

Railway emissions for the ferry service through the Strait are due to the movements of the shunting locomotives used for boarding railway wagons in the trip from the station of Villa San Giovanni to the ferry docks.

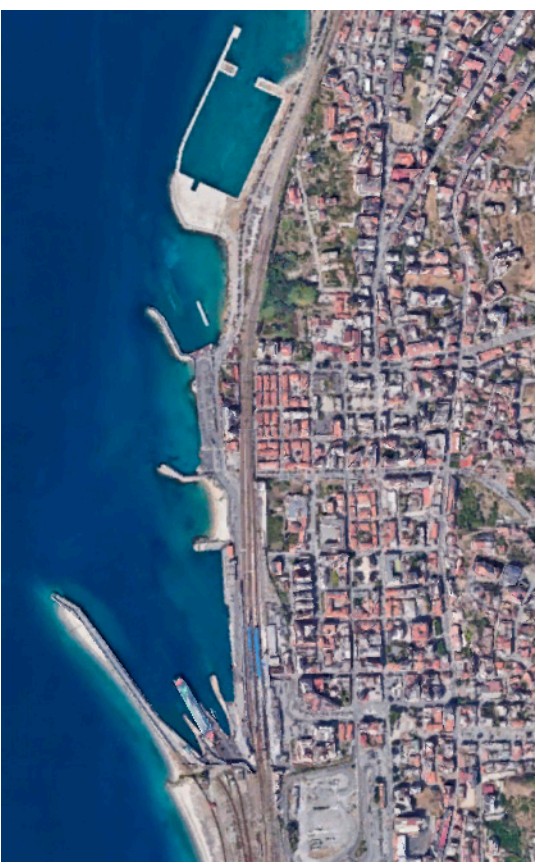

**Figure 13.** Villa San Giovanni port (Source: Map data © 2023 Google).

**Table 21.** Characteristics of engines of ships in service at Villa San Giovanni port.

| Type of Ship | Company | $P_{max}$ (kW) | $P_{aux}/P_{main}$ |
|---|---|---|---|
| Passenger ship | Blu Jet | 4000 | 0.27 |
| Ro-ro ferry | Caronte &Tourist | 8800 | 0.39 |
| | Bluferries | 7900 | 0.39 |
| Railway ferry | R.F.I. | 10,500 | 0.39 |

**Table 22.** Average time for each phase at Villa San Giovanni port.

| Type of Ship | $t_{docking}$ (min) | $t_{hotelling}$ (min) | $t_{departure}$ (min) |
|---|---|---|---|
| Passenger ship | 3 | 10 | 3 |
| Ro-ro ferry (Caronte & Tourist) | 3 | 20 | 3 |
| Ro-ro ferry (Blueferries) | 5 | 20 | 3 |
| Railway ferry | 6 | 60 | 5 |

**Table 23.** Distances Ferry Terminal—Villa San Giovanni motorway junction.

| Terminal | Terminal → Junction $d_v$ (km) | Junction → Terminal $d_v$ (km) |
|---|---|---|
| Blufferies | 2.9 | 6.2 |
| Caronte & Tourist | 3.2 | 6.5 |

## 4. Results and Discussion

The following figures show the annual comparison among maritime and road traffic flows occurring in the different ports of the Strait of Messina in 2019 and 2020.

The analysis of the processed data shows that annual ferry activity in the Strait of Messina area decreased in 2020 in all ports but Tremestieri, with percentages varying from 19% in Messina to 8% in Reggio Calabria. In the port of Tremestieri, there was a 6% increase (Figure 14).

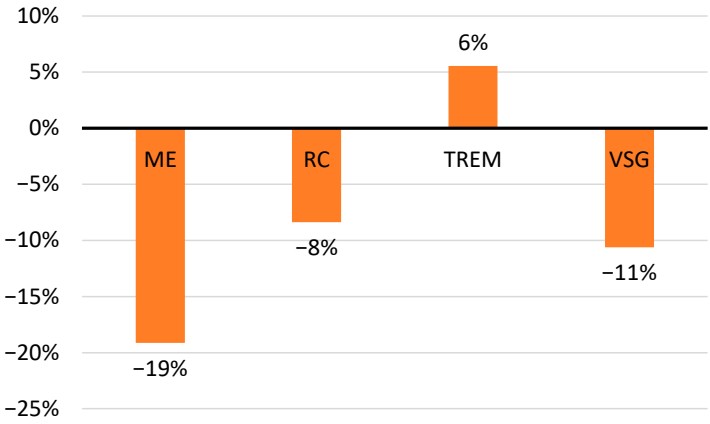

**Figure 14.** Variation of maritime traffic in the Strait of Messina in 2020.

On a monthly basis, the largest year-on-year variation occurred in April (Figure 15), where reductions in ship flows varied from 73% in the port of Messina to 6% in the port of Tremestieri.

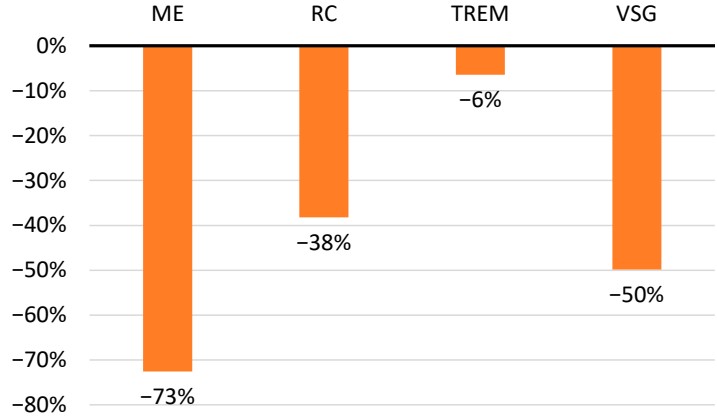

**Figure 15.** Reduction in maritime traffic in the Strait of Messina in April 2020.

The reduction in maritime flows is a direct result of the reduction in vehicular flows caused by the restrictions on mobility in response to the pandemic [65].

With regard to vehicular flows (Figure 16), reductions were recorded in all ports with the exception of the port of Tremestieri, where there was an annual increase due to the transfer of part of the vehicular flow previously directed to the port of Messina. The percentage variations of vehicular flow in the four ports, distinguished in the two rates due to passenger cars and heavy vehicles, is shown in Figure 16.

As a result of the actions taken to limit the spread of the epidemic, car flows between the two sides of the strait were almost zero in April 2020 compared to the previous year (Figure 17). In the same month, there was also a significant reduction of 87% in truck flows at the port of Messina and, to a lesser extent (39%), at the port of Villa San Giovanni, while they remained almost unchanged at the ports of Reggio Calabria and Tremestieri (Figure 17).

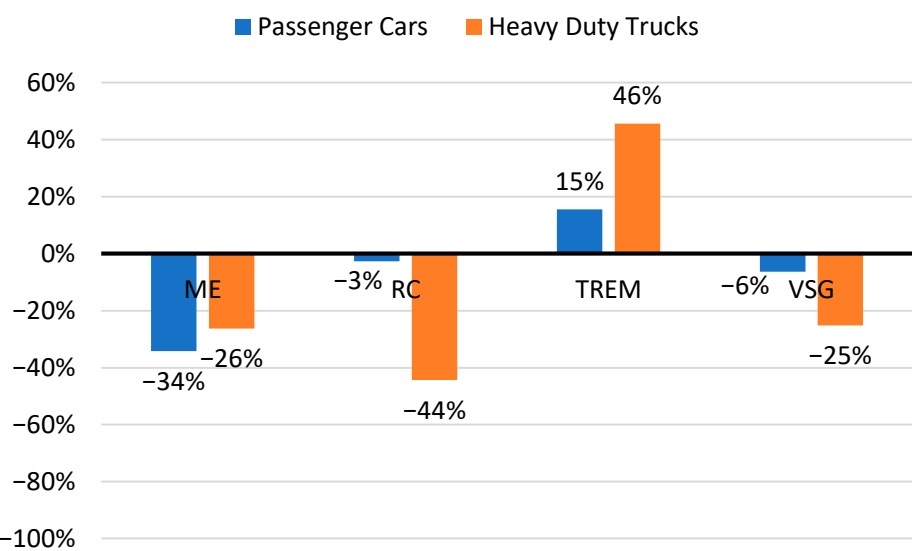

**Figure 16.** Variation of road traffic in the Strait of Messina in 2020.

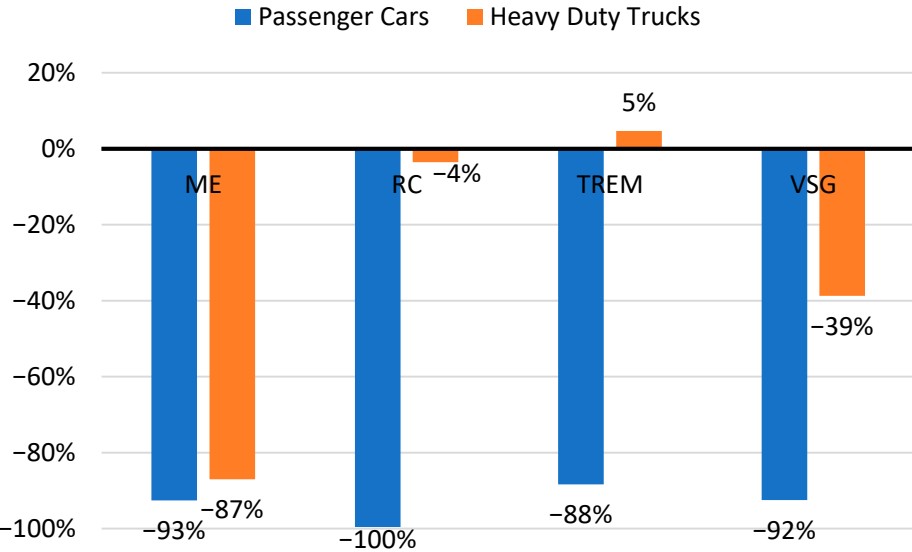

**Figure 17.** Variation of road traffic in the Strait of Messina in April 2020.

The following figures show annual comparisons of emissions that occurred in the different port areas by pollutant source type.

In particular, comparisons due to maritime traffic are shown in Figure 18, distinguishing the contributions due to railway ferries, ro-ro ferries, and passenger ships; the impact of road traffic is shown in Figure 19, for the two components due to cars and heavy vehicles; and the emission of shunting locomotives is shown in Figure 20. At last, total emissions by port area are shown in Figure 21.

In regard to maritime traffic (Figure 18), it can be seen that the highest emissions of pollutants occur in the port of Villa San Giovanni, followed by the port of Messina, with the largest share due to ro-ro ferries.

Villa San Giovanni is also the port with the highest emissions from road traffic, due to it having the longest route between the ferry terminal and the highway junction (Figure 19). Emissions generated by rail traffic occur only in the port of Messina and Villa San Giovanni and, in any case, they are negligible compared to maritime and road emissions (Figure 20).

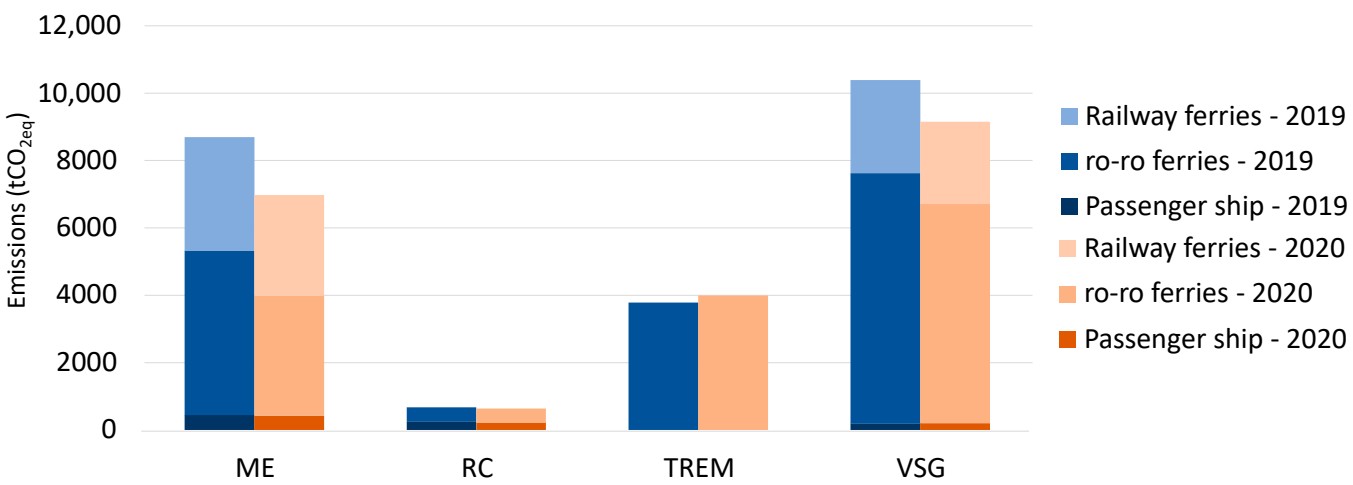

**Figure 18.** Yearly emissions by maritime traffic for each port.

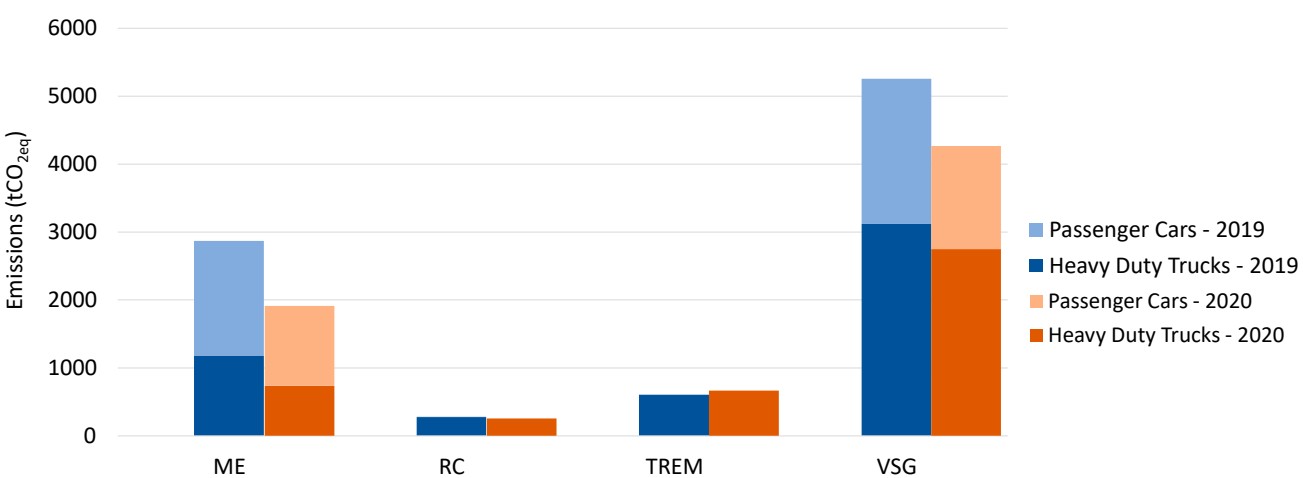

**Figure 19.** Yearly emissions by road traffic for each port.

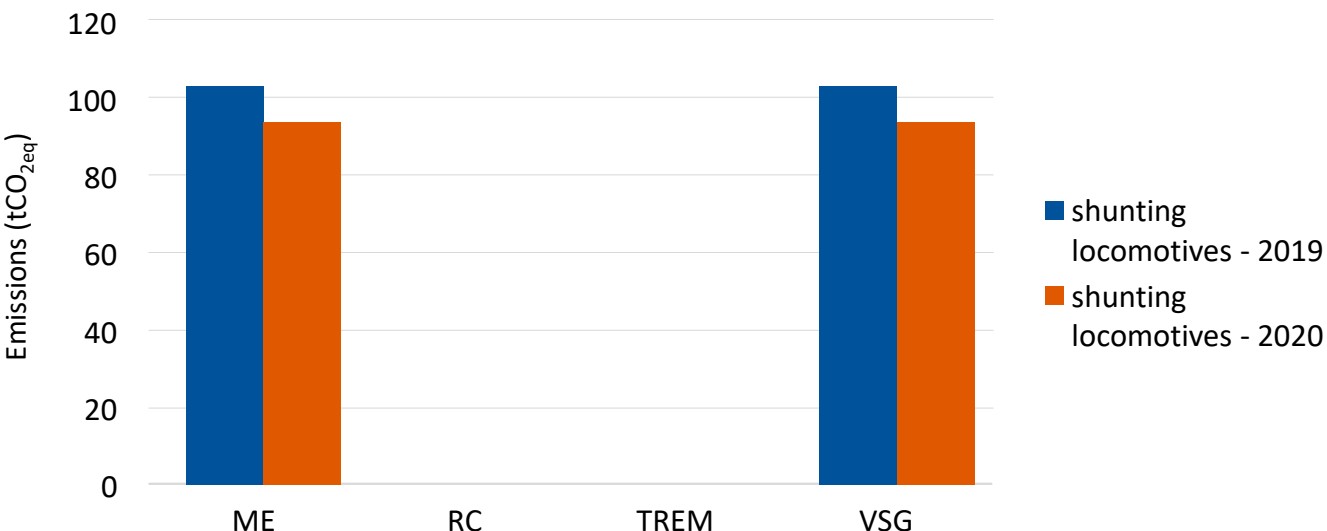

**Figure 20.** Yearly emissions by rail traffic for each port.

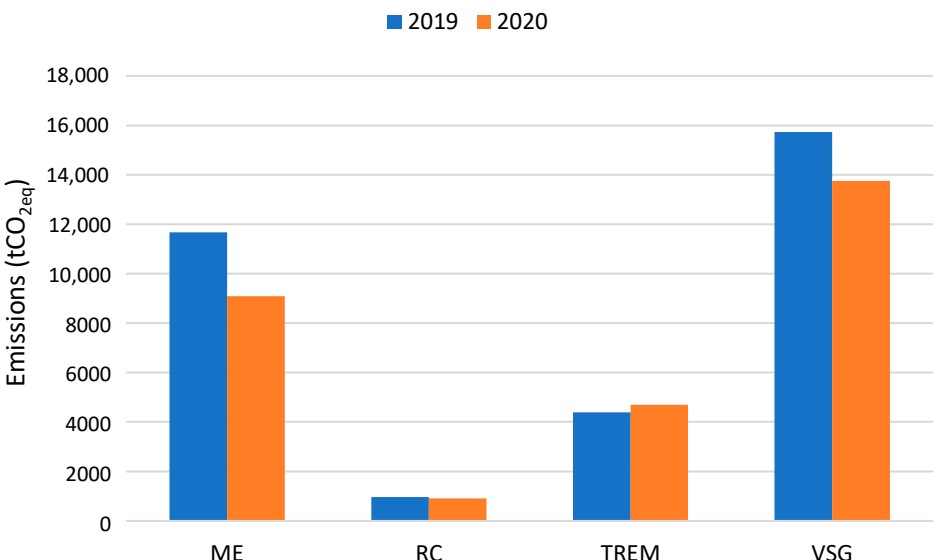

**Figure 21.** Overall pollutant emissions for each port.

With respect to the overall emissions, the highest impact derives from the Villa San Giovanni port (Figure 21).

Starting from the emissions by each port area, the total emissions from ferry activity, divided by pollutant macro-sector, for the two years under consideration are shown in Figure 22.

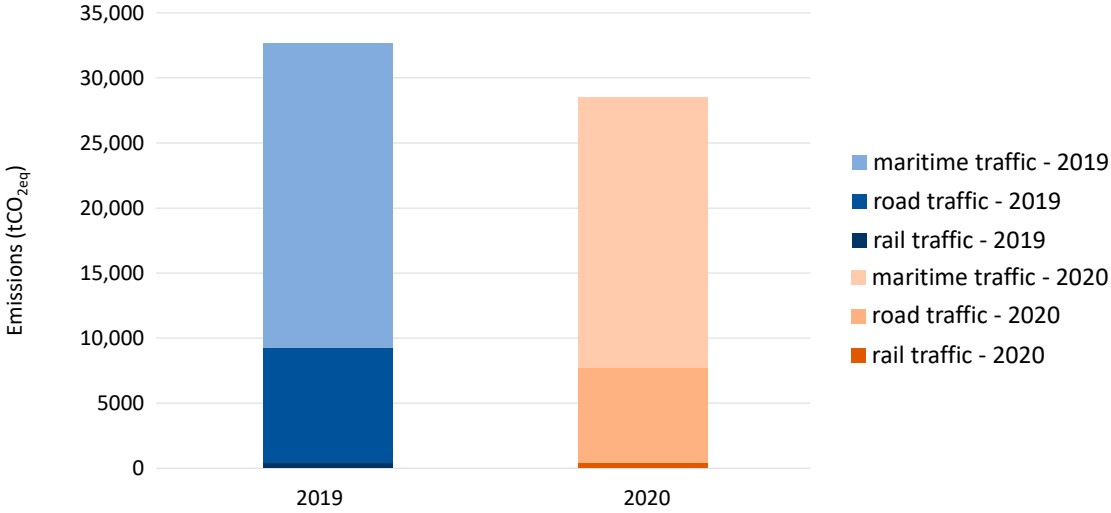

**Figure 22.** Emissions by pollutant macro-sector.

It can be inferred that an overall reduction of 13,2% in $CO_{2eq}$ yearly emission rates was observed, corresponding to an absolute reduction of 4310 $tCO_{2eq}$; the relative reduction is comparable to the results of other studies [66]. It is also noteworthy that the major reduction of 2784 $tCO_{2eq}$ is due to maritime traffic.

Detailed trends of monthly emissions for each port and for each pollutant source are reported in Appendix A.

Variations in $CO_2$ emissions produced annually and in April 2020 due to changes in naval and vehicular flows are shown in Figures 23 and 24, in absolute and percentage values. Emissions from rail traffic are not reported because of their limited magnitude.

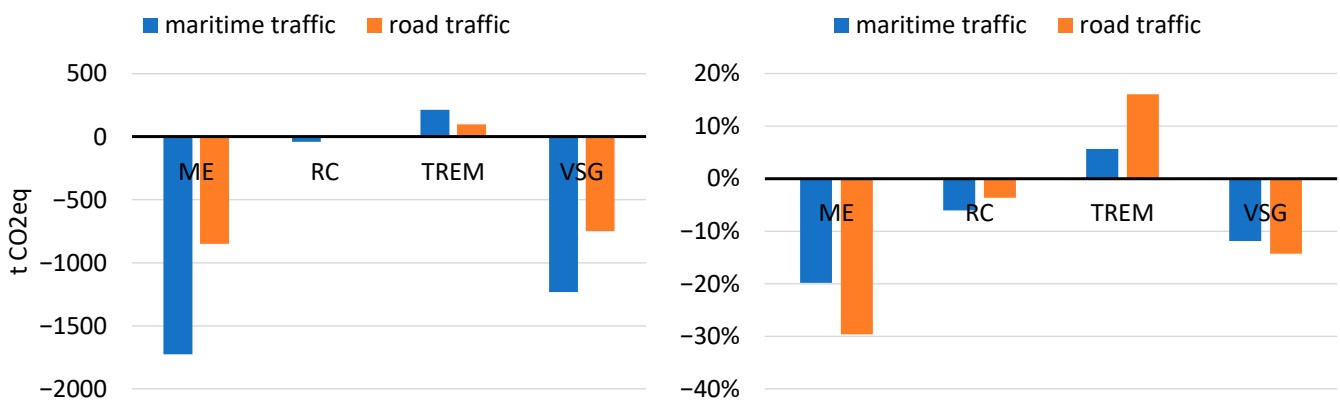

**Figure 23.** Variation of $CO_2$ emissions in 2020.

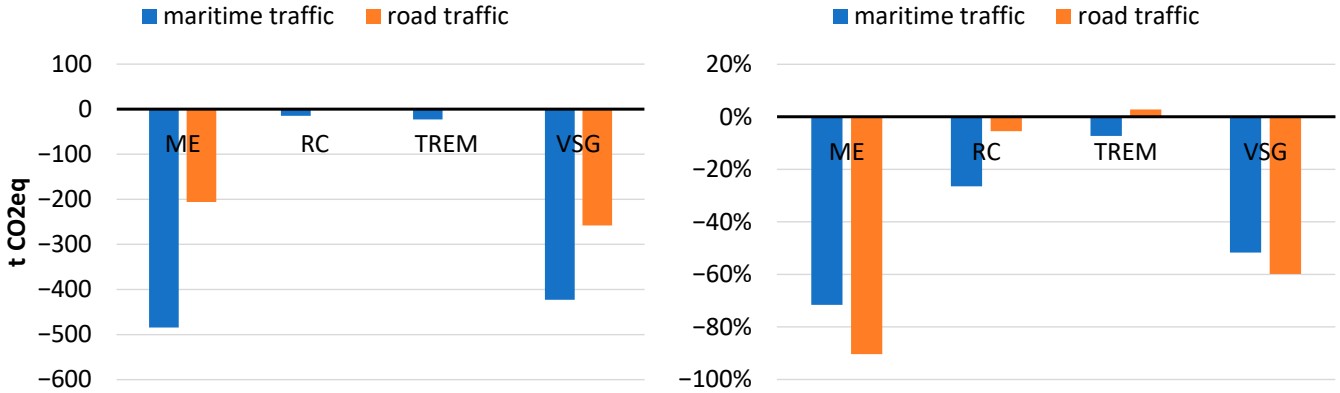

**Figure 24.** Variation of $CO_2$ emissions in April 2020.

It can be seen that although the percentage reduction in emissions from road traffic is greater than the one from ship traffic, in absolute terms this reduction is lower. This is because the most polluting activity is the movement of ships (Figure 22).

Indeed, the almost non-existent passenger car flows in April 2020 had a limited impact because their emission category is less polluting than the others. A similar percentage reduction in heavy vehicles or, even more so, in ship traffic would have produced a more significant reduction in overall emissions. In the perspective of actions to reduce emissions in the area, it is thus appropriate to focus on reducing ship traffic as a priority.

## 5. Conclusions

The spread of the SARS-CoV-2 epidemic has produced a series of restrictive measures that have generally led to a reduction in traffic in the different modes of transport and, as a consequence, to a reduction in atmospheric pollutant emissions.

In this context, an analysis of pollutant emissions was conducted in the Strait of Messina, a southern Italy area characterized by high ferry activities in the routes between the Sicilian and the Calabrian shores, where ferry services involve a total of four ports.

Specifically, owing to both the availability of pre- and post-pandemic mobility data, and the features of the facility, this port area became a noteworthy case study, suited to assess the environmental impact of the various activities here located and to draw conclusions regarding possible future courses of action designed to curb greenhouse gas emissions in these types of settlements.

From the analysis of ship, road, and rail flows, atmospheric $CO_2$ emissions were assessed in 2019, before the virus spread, and in 2020 when the epidemic reached its peak.

As a matter of fact, the restrictions imposed to counteract the spread of COVID-19 regarded non-vital movements and activities; consequently, they provoked an actual scenario of minimal human activity that can be used as a reference for comparison purposes,

with a view to assessing the possible contribution of the investigated facility to climate change mitigation.

The assessments were conducted on a monthly basis in order to single out periods of maximum emission reductions within a year.

Emissions were calculated in a disaggregated basis for the different emission components involved in ferry activities using the EMEP/EEA methodology. In detail, for maritime traffic, emissions attributable to the different types of vessels were evaluated as distinguished into passenger ships, ro-ro ferries and railway ferries, while for vehicular traffic, emissions were evaluated separately for passenger cars and heavy-duty trucks.

The analysis conducted showed that during the pandemic there was a significant reduction in annual traffic in three of the four ports in the Strait area. This has led to an overall reduction in $CO_2$ emissions in the area of about 13%, with peaks of 22% for the port of Messina. This rate of emission reduction is comparable to that achieved in other similar contexts.

In contrast, there was a 7% increase in emissions at the port of Tremestieri because, during the pandemic, some of the traffic previously directed to the port of Messina was shifted to it. In absolute terms, there was an overall reduction of 4310 $tCO_{2eq}$, of which 2582 $tCO_{2eq}$ was from the port of Messina and 1988 $tCO_{2eq}$ from the port of Villa San Giovanni. The largest annual reduction is related to maritime traffic and is 2784 $tCO_{2eq}$.

On a monthly scale, the largest variations occurred in April 2020. For the port of Messina, where the reductions were the greatest, there was a 70% decrease in emissions from maritime traffic and a 90% decrease in emissions from road traffic.

In general, it can be seen that, in percentage terms, the reduction in emissions from road traffic was always greater than that from ship traffic; however, since the most polluting activity as a whole was due to the movement of ships, this fact does not occur in absolute terms.

As a result, it is evident that the almost non-existent passenger car flows in April 2020 had a limited impact on $CO_2$ emissions because their emissions category is less polluting than that of the others. A similar percentage reduction in heavy-duty vehicles or, even more, in ship traffic would produce a more significant reduction in overall emissions. This result may have significant implications with reference to the design of future policies aimed at the development of port areas, which consider the effects on climate-changing emissions.

In conclusion, with a view to designing future actions aimed at reducing pollutant emissions in the area, it is advisable to focus on ships as a priority and only afterwards take measures of the road component.

## 6. Limitation of the Study

Uncertainty assessment is an important part of compiling an emissions inventory and evaluating its evolution over time. Uncertainty in emission estimates is a function of the accuracy of the activity data available to compile the inventory.

The methodological approach for estimating pollutant emissions involves different levels of detail. In the simplest case, Tier 1, the emission inventory is compiled by collecting activity level data and applying appropriate emission factors. Tier 2 and 3 approaches allow for more precise assessments, but use more complex expressions and, more importantly, require more detailed input data.

In this specific case, with reference to available mobility data, Tier 3 was used for ship movements and Tier 1 for road and rail traffic.

## 7. Significance of the Study

Maritime facilities are often the object of the strategies and policies addressing the issue of climate change because they are deemed able to play a pivotal role in reducing greenhouse gas (GHG) emissions worldwide [67–70].

In order to be effective, these strategies need to be based on solid data, figures and analysis, with a view to singling out the best course of action. In this framework, the study

here proposed provides a contribution to the matter; it exploits the effect of the restrictions adopted to counteract the spread of the COVID-19 pandemic to analyse different actual scenarios of traffic in a large port area, composed of four harbours and involving various activities due to ships, ferrying road vehicles, and shunting locomotives. Specifically, owing to both the availability of data and the features of the facility, the port area became a noteworthy case study, suited to assess the environmental impact of the various activities here located and to draw conclusions regarding possible future courses of action designed to curb greenhouse gas emissions in these types of settlements.

In short, the results highlighted in this study could support the design of policies and strategies aimed at mitigating climate change causes.

## 8. Novelty of the Research

The study uses the unprecedented reductions in traffic activity due to the 2020 pandemic to assess the environmental impact of a typical port settlement in terms of both overall emission rates and the share of each involved activity.

As a matter of fact, the restrictions imposed to counteract the spread of the COVID-19 disease regarded non-vital movements and activities; consequently, they provoked an actual scenario of minimal human activity that can be used as a reference for comparison purposes, with a view to assessing the possible contribution of the investigated facility to climate change mitigation. From this perspective, the approach is quite new. In addition, it is also worthy of note that, albeit research analysing single traffic activity is actually available in the literature, there are only a few quantitative studies focusing on port areas as a whole. In this particular frame, the study provide a feasible contribution.

**Author Contributions:** Conceptualization, A.N. and C.M.; methodology, A.N. and C.M.; software, M.F.P. and M.P.; validation, M.F.P. and M.P.; formal analysis, C.M. and M.P.; investigation, C.M. and M.F.P.; data curation, A.N. and M.F.P.; writing—original draft preparation, review and editing, A.N. and C.M.; supervision, A.N. and M.P. All authors have read and agreed to the published version of the manuscript.

**Funding:** This research received no external funding.

**Institutional Review Board Statement:** Not applicable.

**Informed Consent Statement:** Not applicable.

**Data Availability Statement:** Not applicable.

**Conflicts of Interest:** The authors declare no conflict of interest.

## Appendix A

In this appendix, for each port, the monthly emissions in the years 2019 and 2020 are showed and compared; they are subdivided firstly into macro pollutant source (maritime, road and rail traffic) and then into specific emissive component (type of ship, type of road vehicle).

*Appendix A.1. Emissions by Maritime Traffic*

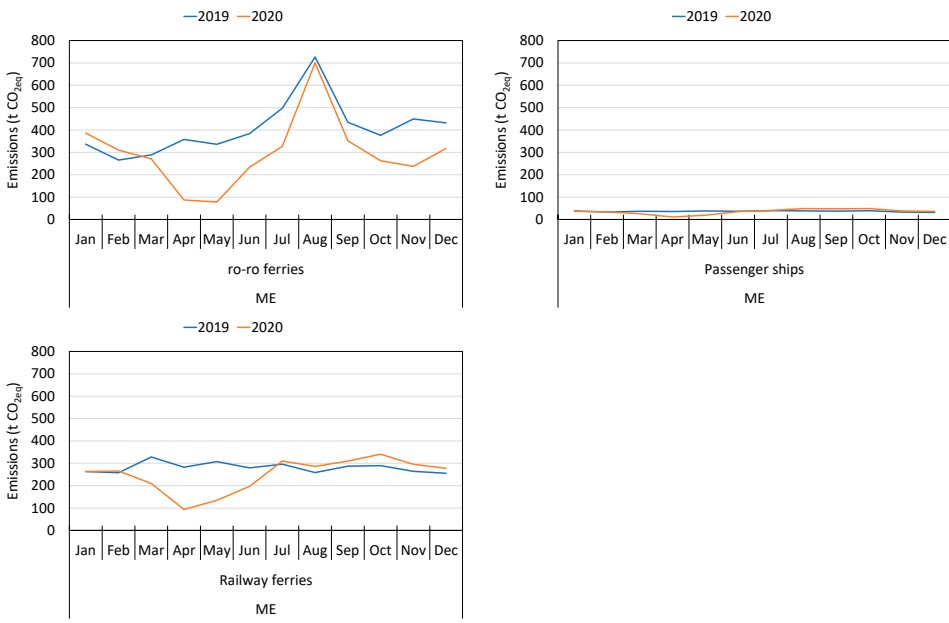

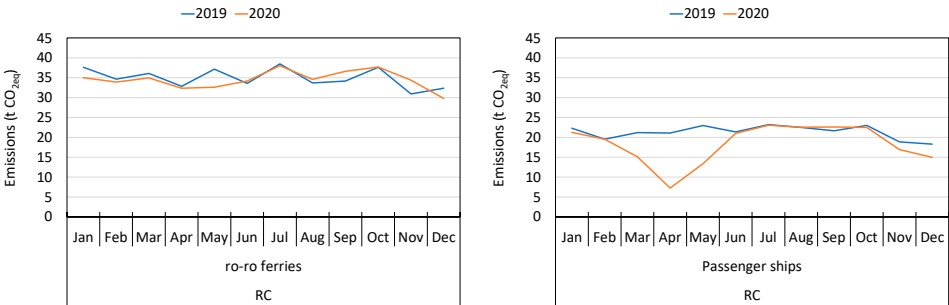

**Figure A1.** Monthly pollutant emissions (docking + hotelling + departure) at Messina port.

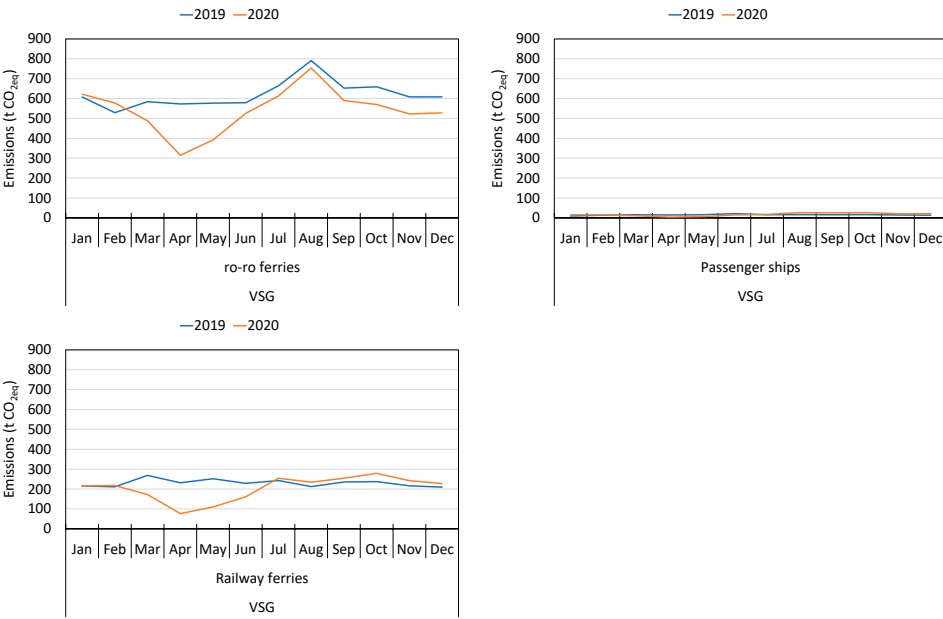

**Figure A2.** Monthly pollutant emissions (docking + hotelling + departure) at Reggio Calabria port.

**Figure A3.** Monthly pollutant emissions (docking + hotelling + departure) at Villa San Giovanni port.

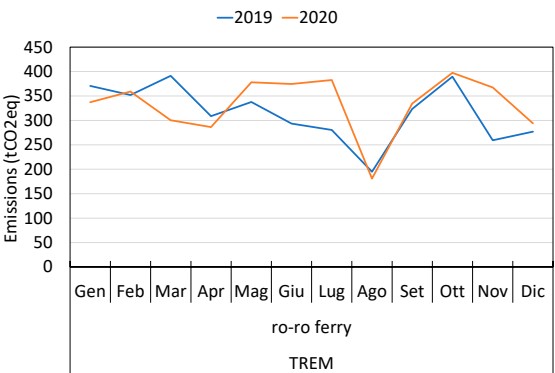

**Figure A4.** Monthly pollutant emissions (docking + hotelling + departure) at Tremestieri port.

*Appendix A.2. Emissions by Road Traffic*

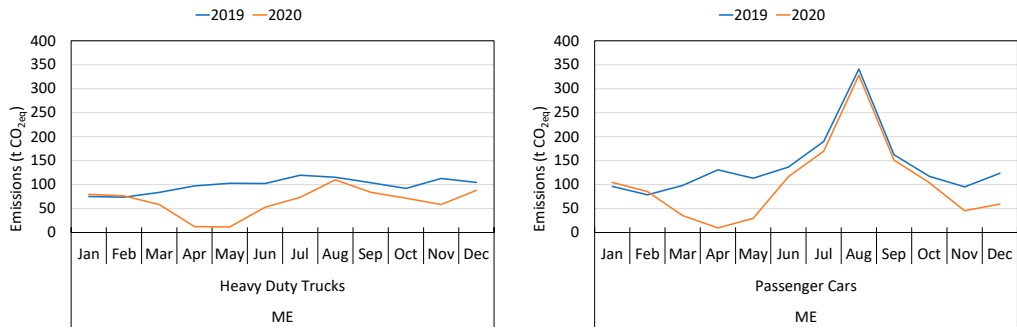

**Figure A5.** Monthly pollutant emissions (embarking + disembarking) at Messina port.

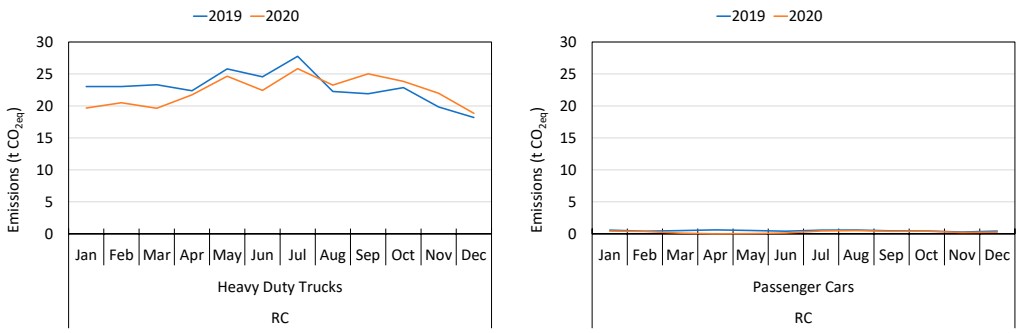

**Figure A6.** Monthly pollutant emissions (embarking + disembarking) at Reggio Calabria port.

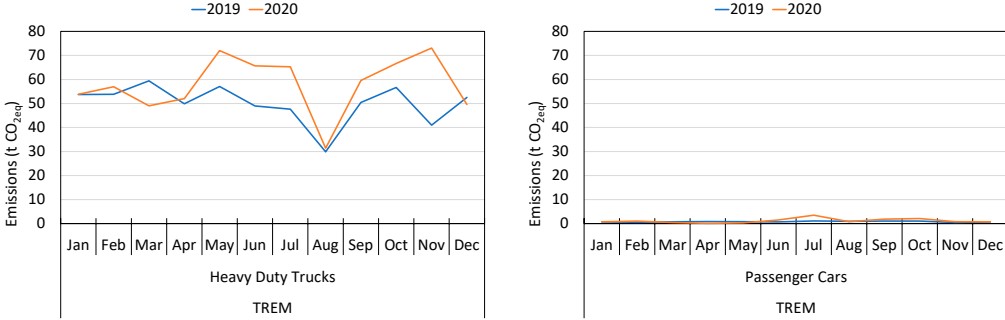

**Figure A7.** Monthly pollutant emissions (embarking + disembarking) at Tremestieri port.

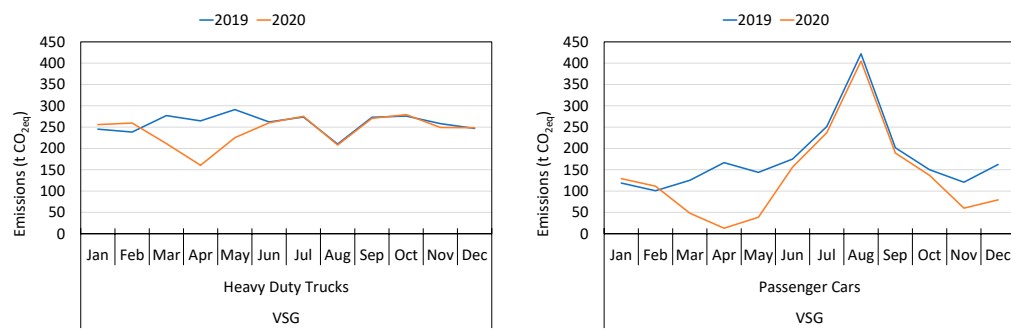

**Figure A8.** Monthly pollutant emissions (embarking + disembarking) at Villa San Giovanni port.

*Appendix A.3. Emissions by Rail Traffic*

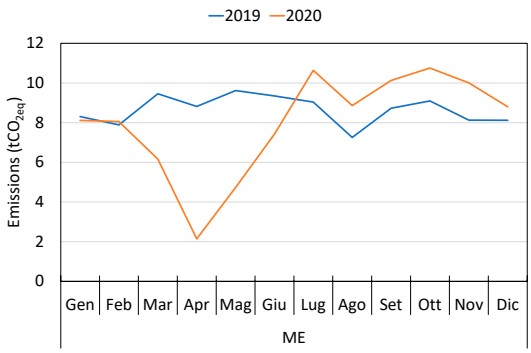

**Figure A9.** Monthly railway pollutant emissions at Messina port.

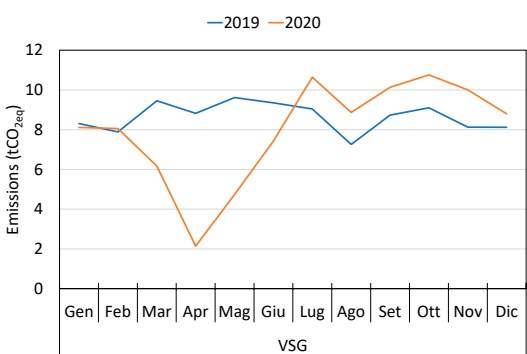

**Figure A10.** Monthly railway pollutant emissions at Villa San Giovanni port.

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
