# Peer review of "Effects of the SARS-CoV-2 Pandemic on CO2 Emissions in the Port Areas of the Strait of Messina"

_sustainability, doi:10.3390/su15129587_

Round 1

Reviewer 1 Report

interesting paper assessing the effect of pandemic to the ferry traffic and CO2 emission. The paper is well organized and supported, here are my comments:

Satellite images should be given source/citation

Conclusion is weak as the contribution of research is missing

Reducation of traffic is correlated with reduction of pollutants. any significant or different findings should be highlighted in the paper.

What is the limitations of method and conclusion? more years of data and comparison?

Minor editing of English language required

Reviewer 2 Report

The paper focuses on the impact of the SARS-CoV-2 pandemic on carbon dioxide (CO2) emissions in the Strait of Messina port areas. The researchers highlight the significant ship flow in the strait, particularly ferry services between Messina and Tremestieri in Sicily, Reggio Calabria, and Villa San Giovanni in Calabria. The pandemic-related restrictions, aimed at limiting the spread of the virus, reduced ferry activities and vehicle flows during the epidemic emergency. Consequently, there was a decrease in atmospheric pollutant emissions. The study aims to assess the effect of the implemented restrictions on CO2 emissions by comparing the total emissions from ferry activities in two consecutive years: 2019 (pre-pandemic) and 2020 (peak of the epidemic). The researchers utilized the EMEP/EEA "Air pollutant emission inventory guidebook 2019" emission estimation methodology to analyze emissions originating from ships, ferrying road vehicles, and shunting locomotives used for moving rail cars.

In summary, the study demonstrates the significant reduction in CO2 emissions in the port areas of the Strait of Messina during the SARS-CoV-2 pandemic, emphasizing the need to prioritize measures to reduce emissions from ships in future sustainability efforts.

It would be beneficial to include a section discussing the potential limitations of the study. Addressing factors such as data availability, assumptions made in the emission estimation methodology, or uncertainties in the analysis would strengthen the paper and provide a more comprehensive review of the research.

Moreover, some research like 

https://doi.org/10.1016/j.jtte.2020.04.006

is interesting in regard to the new view of the emission concept.

It's important to ensure that the paper is free of any grammatical or typographical errors to enhance the overall readability and professionalism of the manuscript.

Reviewer 3 Report

This paper explains the effect of SARS-CoV-2 pandemic on CO2 in the port areas of the Strait of Messina. Authors have analysed the 2 years’ data from 2019 to 2020 during pandemic period was at peak sine most of the vehicles and ferry are not fully in operation. They compared the CO2 data during the study period with EMEP/EEA standards and analysed the deviation.  Authors have critically reviewed the literatures related to CO2 studies and it effect during the pandemic period. The manuscript is prepared well and suitable for publication in the Sustainability journal of MDPI. It is well organized and may be accepted after major changes incorporated in the manuscript.

  • There are no numerical values (such as standards or ranges, sensor quality, or calibration) in the abstract and conclusion, Include numerical values in the manuscript
  • The objectives and novelty of the research not properly described in the manuscript
  • Demographic and Socioeconomic condition of the study area is required along with other pollution parameters status during, pre and post pandemic condition is required.
  • Pre and post pandemic data of traffic is not discussed in the manuscript, then how the author can compare its effect during peak pandemic
  • There is lacking of sufficient discussion for many tables and figures in the manuscript and its relation the topic of the research also earlier research works. Kindly include it.
  • Authors have collected and presented voluminous data without proper analysis and discussions
  • Latest literature may be included and compared as post pandemic analysis on traffic condition and CO2
  • Complete modification of manuscript required in connection with figure and tables.

This paper explains the effect of SARS-CoV-2 pandemic on CO2 in the port areas of the Strait of Messina. Authors have analysed the 2 years’ data from 2019 to 2020 during pandemic period was at peak sine most of the vehicles and ferry are not fully in operation. They compared the CO2 data during the study period with EMEP/EEA standards and analysed the deviation.  Authors have critically reviewed the literatures related to CO2 studies and it effect during the pandemic period. The manuscript is prepared well and suitable for publication in the Sustainability journal of MDPI. It is well organized and may be accepted after major changes incorporated in the manuscript.

  • There are no numerical values (such as standards or ranges, sensor quality, or calibration) in the abstract and conclusion, Include numerical values in the manuscript
  • The objectives and novelty of the research not properly described in the manuscript
  • Demographic and Socioeconomic condition of the study area is required along with other pollution parameters status during, pre and post pandemic condition is required.
  • Pre and post pandemic data of traffic is not discussed in the manuscript, then how the author can compare its effect during peak pandemic
  • There is lacking of sufficient discussion for many tables and figures in the manuscript and its relation the topic of the research also earlier research works. Kindly include it.
  • Authors have collected and presented voluminous data without proper analysis and discussions
  • Latest literature may be included and compared as post pandemic analysis on traffic condition and CO2
  • Complete modification of manuscript required in connection with figure and tables.

Reviewer 4 Report

The manuscript titled "Effects of the SARS-CoV-2 pandemic on CO2 emissions in the port areas of the Strait of Messina" intends to test a feasible procedure to assess the impact of port areas in terms of greenhouse gas emissions generated by the various activities. The study area is the port system of the Strait of Messina, a southern Italy area characterized by high ferry activities. The selected port system is composed of 4 harbours characterized by different dimensions and activities: Messina, Tremestieri, Villa San Giovanni and Reggio Calabria.

The research is original; it could be characterized as novel and, in my opinion, important to the field, it also has an almost appropriate structure, and the language has been used well. In the meanwhile, the manuscript has a short length (about 5,900 words) and it is almost comprehensive. The figures (34), the tables (23) and the equations (5) make the paper reflect well to the reader. For this reason, paper has a "diversity look", not only numbers, not only words.

It is advised to revise figures, compare them, or use an appendix or present some figures with tables. The total number of tables and figures is excessively big.

The title, I think, is all right. The abstract did not reflect well the findings of this study. Please revise the abstract of the manuscript and do not forget abstract need to encourage readers to download the paper. The Abstract needs further work. It is not clear. Abstracts should indicate the research problem/purpose of the research, provide some indication of the design/methodology/approach taken, the findings of the research and its originality/value in terms of its contribution to the international literature. The abstract has a nice length (about 170 words). Please, revise the abstract, it must be up to 200 words long, [see: Instructions for Authors / Manuscript Submission Overview / Accepted File Formats - (https://www.mdpi.com/journal/sustainability/instructions#submission or https://www.mdpi.com/files/word-templates/sustainability-template.dot)].

The introduction is effective, clear, and well organized but it wasn’t introduced and put into perspective what research is negotiating. Moreover, it does not contain a clear formulation and description of the research problem. Please insert a clear description and justification of the problem the article deals with. Your literature research should be critical and more informed, rather than listing previous research. This section requires significant improvement. Because descriptive statistics are used, it is advisable to state the research hypotheses and perform hypothesis testing.

For the Methodology chapter, the research conduct has been tested in several areas of the world, with comparable results and will probably be tested in others. Appropriate references to the methodology included in the already published bibliography but you can put more references, from all over the world. Do not forget, the journal “Sustainability” is international.

The results section is good. The argument flows and is reinforced through the justification of the way elements are interpreted. But the same does not apply to the Discussion and Conclusion. Both sections should be consistent in terms of Proposal, Problem statement, Results, and of course, future work. Your conclusion section does not do justice to your work. Make your key contributions, arguments, and findings clearer. You must refer to the literature and previous studies in your discussion section.

Please revise the manuscript and include more references which already exist in the bibliography. I would be much more satisfied if the number of references was slightly higher (about 70 - 80 references) and I would appreciate it if it also included data from all the word (Asia, America, Europe, or Australia). In this way it is documented that a method that is tested in a place with its own characteristics can be implemented in other places around the world. Your references are only from books, working papers and reports, you have not used journal articles, why?

Please revise the references of the manuscript and include references which are already exists in bibliography. References must have an appropriate style, for this reason I would be good to change [see: Instructions for Authors / Manuscript Preparation / Back Matter / References: - (https://www.mdpi.com/journal/sustainability/instructions or https://www.mdpi.com/authors/references)]. Do not forget, DOI numbers (Digital Object Identifier) are not mandatory but highly encouraged and make the review easier.

More discussion is needed, comparing the results of this work related to attributes with those of other studies. I believe that the conclusions section or discussion should also include the main limitations of this study and incorporate possible policy implications. I think something more should be said about practical implications.

Also use the appropriate research manuscript sections: Introduction, Materials and Methods, Results, Discussion and Conclusions, as journal wants (https://www.mdpi.com/journal/sustainability/instructions). You must organize your paper. You have many subchapters, please reduce them.

In line 458 Patents: this section is not mandatory but may be added if there are patents resulting from the work reported in the manuscript. You can delete it.

Please fill in the subchapters accordingly as: Author Contributions, Funding, Institutional Review Board Statement, Informed Consent Statement, Data Availability Statement, Acknowledgments and Conflicts of Interest, according to the instructions of the International Journal Sustainability [see: Instructions for Authors / Manuscript Preparation/ Back Matter - (https://www.mdpi.com/journal/sustainability/instructions#submission or https://www.mdpi.com/files/word-templates/ sustainability-template.dot)].

For Author Contributions, the following statements should be used "Conceptualization, X.X. and Y.Y.; Methodology, X.X.; Software, X.X.; Validation, X.X., Y.Y. and Z.Z.; Formal Analysis, X.X.; Investigation, X.X.; Resources, X.X.; Data Curation, X.X.; Writing – Original Draft Preparation, X.X.; Writing – Review & Editing, X.X.; Visualization, X.X.; Supervision, X.X.; Project Administration, X.X.; Funding Acquisition, Y.Y.”

Minor editing of English language required

Round 2

Reviewer 3 Report

  • The authors included numerical results in the abstract and conclusion.
  • Regarding the objectives of the research, they are not mentioned in the manuscript.
  • Demographic and related data included
  • Pre- and post-pandemic data on traffic included
  • Still lacks an explanation for figures (example: figures 4, 5, and 6). It is only given as shown.
  • The latest literature is included.
  • Modified as suggested in my earlier review.

Author Response

  • Regarding the objectives of the research, they are not mentioned in the manuscript.
    • The research objectives were highlighted in red in the introduction from row 75 to row 98; they are also specified in sections 7. Significance of Study and 8. Novelty of the Research.

  • Still lacks an explanation for figures (example: figures 4, 5, and 6). It is only given as shown.
    • Reference to the figures has been included and is highlighted in red in the text.

Reviewer 4 Report

The manuscript titled "Effects of the SARS-CoV-2 pandemic on CO2 emissions in the port areas of the Strait of Messina" intends to test a feasible procedure to assess the impact of port areas in terms of greenhouse gas emissions generated by the various activities. The study area is the port system of the Strait of Messina, a southern Italy area characterized by high ferry activities. The selected port system is composed of 4 harbours characterized by different dimensions and activities: Messina, Tremestieri, Villa San Giovanni and Reggio Calabria.

The manuscript has been revised according to the first review comments. The authors carefully studied the comments and revised the manuscript by considering all the last comments. The comments are responded to the new manuscript.

Conclusions and discussion are better than the previous one, they have general logic and on justification of interpretations as the author’s attribute.

In general, the manuscript is completely different from the previous one, since all the comments of the previous review have been revised. I would be much more satisfied if the number of references was slightly higher (about 70 - 80 references).

I believe the revised manuscript has been improved carefully and I hope the desired level of Sustainability can be reached.

Minor editing of English language required.

Author Response

In general, the manuscript is completely different from the previous one, since all the comments of the previous review have been revised. I would be much more satisfied if the number of references was slightly higher (about 70 - 80 references).

Other references were added. Now their number is 70.